# A temporally dynamic *Foxp3* autoregulatory transcriptional circuit controls the effector Treg programme

David Bending[1,†] iD, Alina Paduraru[1], Catherine B Ducker[1], Paz Prieto Martín[1], Tessa Crompton[2] & Masahiro Ono[1,*] iD

## Abstract

Regulatory T cells (Treg) are negative regulators of the immune response; however, it is poorly understood whether and how *Foxp3* transcription is induced and regulated in the periphery during T-cell responses. Using *Foxp3*-Timer of cell kinetics and activity (Tocky) mice, which report real-time *Foxp3* expression, we show that the flux of new *Foxp3* expressors and the rate of *Foxp3* transcription are increased during inflammation. These persistent dynamics of *Foxp3* transcription determine the effector Treg programme and are dependent on a Foxp3 autoregulatory transcriptional circuit. Persistent *Foxp3* transcriptional activity controls the expression of coinhibitory molecules, including CTLA-4 and effector Treg signature genes. Using RNA-seq, we identify two groups of surface proteins based on their relationship to the temporal dynamics of *Foxp3* transcription, and we show proof of principle for the manipulation of *Foxp3* dynamics by immunotherapy: new *Foxp3* flux is promoted by anti-TNFRII antibody, and high-frequency *Foxp3* expressors are targeted by anti-OX40 antibody. Collectively, our study dissects time-dependent mechanisms behind Foxp3-driven T-cell regulation and establishes the *Foxp3*-Tocky system as a tool to investigate the mechanisms behind T-cell immunotherapies.

**Keywords** Foxp3; immunotherapy; Tocky; transcriptional dynamics; Treg
**Subject Categories** Immunology; Transcription
**The EMBO Journal (2018) 37: e99013**

## Introduction

Upon antigen recognition through the T-cell receptor (TCR), T cells express interleukin(IL)-2 and CD25 (IL-2 receptor alpha chain), which together promote T-cell activation, proliferation, and differentiation (Shimizu *et al*, 1986; Gaffen, 2001). Intriguingly,

CD25-expressing T cells from healthy animals are markedly enriched with regulatory T cells (Treg) that express the transcription factor Foxp3 (Fontenot *et al*, 2003; Hori *et al*, 2003). Foxp3 expression is a major determinant of Treg phenotype and function, and Foxp3 interacts with transcription factor complexes, such as those involving NFAT and Runx1, to repress IL-2 transcription and convert the effector mechanisms in T cells into a suppressive one (Wu *et al*, 2006; Ono *et al*, 2007; Rudra *et al*, 2012). Treg have activated phenotypes, and upon TCR signals, Treg suppress the activities of conventional T cells (Shevach, 2000). TCR signalling is the major regulator of Treg differentiation in the thymus, as T cells that have received strong TCR signals preferentially express CD25 and Foxp3 and differentiate into Treg (Hsieh *et al*, 2012). Additionally, costimulatory receptors augment TCR signal-dependent Foxp3 and CD25 expression (Tai *et al*, 2005; Mahmud *et al*, 2014). In the periphery, strong TCR signals further differentiate Treg into "effector Treg", showing enhanced suppressive function (Rosenblum *et al*, 2016).

Accumulating evidence indicates that Foxp3 expression is dynamically controlled in Treg and non-Treg. TCR stimulation induces Foxp3 expression in human (Tran *et al*, 2007) and mouse T cells (Miyao *et al*, 2012) *in vitro*. Studies using T-cell receptor (TCR) transgenic systems have shown that Foxp3 expression is induced in non-Treg in some inflammatory conditions *in vivo* (Curotto de Lafaille *et al*, 2008). Although such induced Foxp3 expression is often dismissed as "transient expression", the dynamic induction of Foxp3 expression may have functional roles during T-cell responses if this reactive Foxp3 expression occurs in activated polyclonal T cells during inflammation *in vivo* (Ono & Tanaka, 2016). In addition, Foxp3 expression can be dynamically downregulated in Treg. Fate-mapping experiments showed that, while most of thymus-derived Foxp3[+] T cells stably express Foxp3, some Foxp3[+] cells downregulate Foxp3 to become ex-Foxp3 cells in the periphery, joining the memory-phenotype T-cell pool (Miyao *et al*, 2012). PD-1 KO mice with a partial Foxp3 insufficiency lead to generation of ex-Foxp3 effector T cells (Zhang *et al*, 2016), indicating that the mechanism of T-cell activation is involved in the dynamic regulation of *Foxp3* transcription. These findings lead to the hypothesis that Foxp3 acts

1   Department of Life Sciences, Faculty of Natural Sciences, Imperial College London, London, UK
2   UCL Great Ormond Street Institute of Child Health, London, UK
    *Corresponding author. Tel: +44 20 7594 3895; E-mail: m.ono@imperial.ac.uk
    †Present address: Institute of Immunology and Immunotherapy, College of Medical and Dental Sciences, University of Birmingham, Birmingham, UK

as a cell-intrinsic and transcellular negative feedback regulator for T-cell activation among self-reactive T-cell repertoires (Ono & Tanaka, 2016), challenging the thymus-central view of Treg-mediated immune regulation.

The key question is whether and how frequently activation of new *Foxp3* transcription is induced in non-Treg cells in physiological conditions, and how *Foxp3* transcription is sustained in existing Treg during the immune response. Since the death rate of Treg and other T cells is difficult to determine experimentally, the relative proportions of Foxp3$^+$ and Foxp3$^-$ cells in steady-state conditions may not reflect the probability of new *Foxp3* induction in individual T cells, especially when T cells are expanding and dying during the immune response. Furthermore, human studies show that the level of Foxp3 expression may determine the functional state of Treg: the higher Foxp3 expression is, the more suppressive Treg are (Miyara *et al*, 2009; Fujii *et al*, 2016). Thus, it is fundamental to investigate the temporal dynamics of *Foxp3* transcription over time in individual T cells *in vivo*, but this has been technically difficult to do to date.

Here, we use our novel Timer of cell kinetics and activity (Tocky) system to reveal the time and frequency of *Foxp3* transcription during peripheral immune responses (Bending *et al*, 2018). In the *Foxp3*-Tocky system, the transcriptional activity of the *Foxp3* gene is reported by Fluorescent Timer protein, the emission spectrum of which spontaneously changes from Blue to Red fluorescence after translation (Subach *et al*, 2009). We show that during inflammation, the flux of *Foxp3*$^-$ to *Foxp3*$^+$ T cells dramatically increases within the periphery. In addition, we demonstrate that the real-time frequency of *Foxp3* transcription determines effector Treg differentiation. Thus, we provide experimental evidence that *Foxp3* expression is dynamically regulated in Treg and non-Treg during inflammation *in vivo*, providing fresh insight into Foxp3-driven T-cell regulation.

# Results

### Timer-Blue fluorescence reports real-time *Foxp3* transcription

Fluorescent Timer protein (Timer) is an mCherry mutant (precisely FT-Fast), and when translated, the chromophore of Timer is an unstable blue form, which spontaneously and irreversibly matures to become a stable red form (Subach *et al*, 2009). We hypothesised that Timer-Blue fluorescence in *Foxp3*-Tocky mice reports the real-time transcriptional activities of the *Foxp3* gene. To determine the relationships between *Timer* mRNA expression and endogenous *Foxp3* transcripts, we performed an RNA degradation assay using actinomycin D. After actinomycin D treatment, the transcripts of *Timer*, *Foxp3* and an unrelated mRNA species, *Hprt*, exhibited similar half-lives (1.14, 1.46 and 1.73 h, respectively, Fig 1A). This indicates that *Timer* transcripts are well correlated to *Foxp3* ones in *Foxp3*-Tocky T cells, in which *Timer* transcripts report the transcriptional activity of the *Foxp3* gene (Bending *et al*, 2018). Next, we analysed the half-life of Timer-Blue fluorescence in activated T cells *in vitro* using a short-term treatment with cycloheximide (CHX) to inhibit new protein synthesis. While a previous study estimated the maturation half-life of Timer-Blue to be 7.1 h, using purified Timer proteins and by fitting data to a pharmacological kinetic model (Subach *et al*, 2009), here we aimed to experimentally determine the half-life of Timer-Blue fluorescence by

flow cytometric analysis of *Foxp3*-Tocky T cells. Upon CHX treatment, flow cytometric plots showed that Foxp3-Timer-Blue fluorescence rapidly decayed, while Timer-Red fluorescence was more stable (Fig 1B). The half-life of Timer-Blue fluorescence was estimated to be 4.1 h (Fig 1C), which was in keeping with measurements of Timer-Blue using the *Nr4a3*-Tocky system (Bending *et al*, 2018). In contrast, Timer-Red fluorescence gradually increased (Fig 1D), due to the accumulation of matured proteins. In addition, we investigated the decay of Timer-Red fluorescence by sorting pure Timer-Red$^+$ T cells from *Nr4a3*-Tocky mice (Bending *et al*, 2018), in which Timer-Blue proteins are only induced in response to TCR stimulation, and cultured them in the absence of TCR stimuli (Fig 1E). These data showed that Timer-Red fluorescence was relatively long-lived and decayed with a half-life of approximately 122 h (Fig 1F). The short half-life of Timer-Blue fluorescence indicates that Timer-Blue fluorescence reports real-time *Foxp3* transcripts, while Timer-Red fluorescence captures the cumulative activity of *Foxp3* transcription over a period of 5 days.

### Thymic CD4-single-positive cells have higher rates of new *Foxp3* transcription compared to splenic CD4$^+$ T cells in neonatal mice

In neonatal mice, Foxp3$^+$ T cells are actively produced in the thymus (Dujardin *et al*, 2004). In *Foxp3*-Tocky mice, thymic CD4-single-positive cells showed a higher percentage of Blue$^+$ cells in Timer$^+$ cells than splenic CD4$^+$ cells (i.e. 95 and 63% of all Timer$^+$ cells were Blue$^+$, Fig 2A).

Timer fluorescence data can be quantitatively analysed by the Timer data analysis for the Tocky system (Bending *et al*, 2018). Briefly, Timer fluorescence data are subjected to data pre-processing, including thresholding and Blue-Red normalisation, and subsequently transformed by a trigonometric function to calculate the angle from the Blue axis (Timer-Angle) in the Blue-Red plane (Bending *et al*, 2018; i.e. to use the polar coordination; Fig 2B). Since the maturation of Timer proteins is unidirectional from Blue to Red, Timer expression starts from Blue$^+$Red$^-$ (the New locus), these New cells mature and acquire Red fluorescence within 4 h (Fig 1C and D). When *Timer* transcription persists, cells eventually reach a balanced steady state for Blue and Red fluorescence and accumulate in Blue$^+$Red$^+$ Persistent locus around 45° degree from the normalised Blue axis. When *Timer* transcription is arrested, cells lose Blue fluorescence and stay in the Blue$^-$Red$^+$ Arrested locus while Red proteins decay with half-life of 5 days (Fig 1F). Cells in the Arrested locus can however immediately acquire Blue fluorescence again when they re-initiate *Foxp3* transcription (Fig 2B), indicating that the Timer-Angle between Persistent–Arrested loci represents the recent frequency of *Foxp3* transcriptional activity (Bending *et al*, 2018).

Timer-Blue expression levels in Timer$^+$ cells were much higher in the thymus than in the spleen, while Timer-Red expression levels were higher in the spleen (Fig 2C). These data indicate that the rate of *Foxp3* transcription is higher in the thymus than the spleen, while splenic Foxp3$^+$ cells have transcribed the *Foxp3* gene for a longer time on average than thymic Foxp3$^+$ cells. These results thus further confirm that *Foxp3*-Tocky captures real-time *Foxp3* transcription by Timer-Blue fluorescence and its history and cumulative activity by Timer-Red *in vivo*. The mean Timer-Angle of 10-day-old (D10) neonatal thymus is ~25°, while that of the age-matched spleen is ~75°, indicating that the thymus is enriched with the T cells that

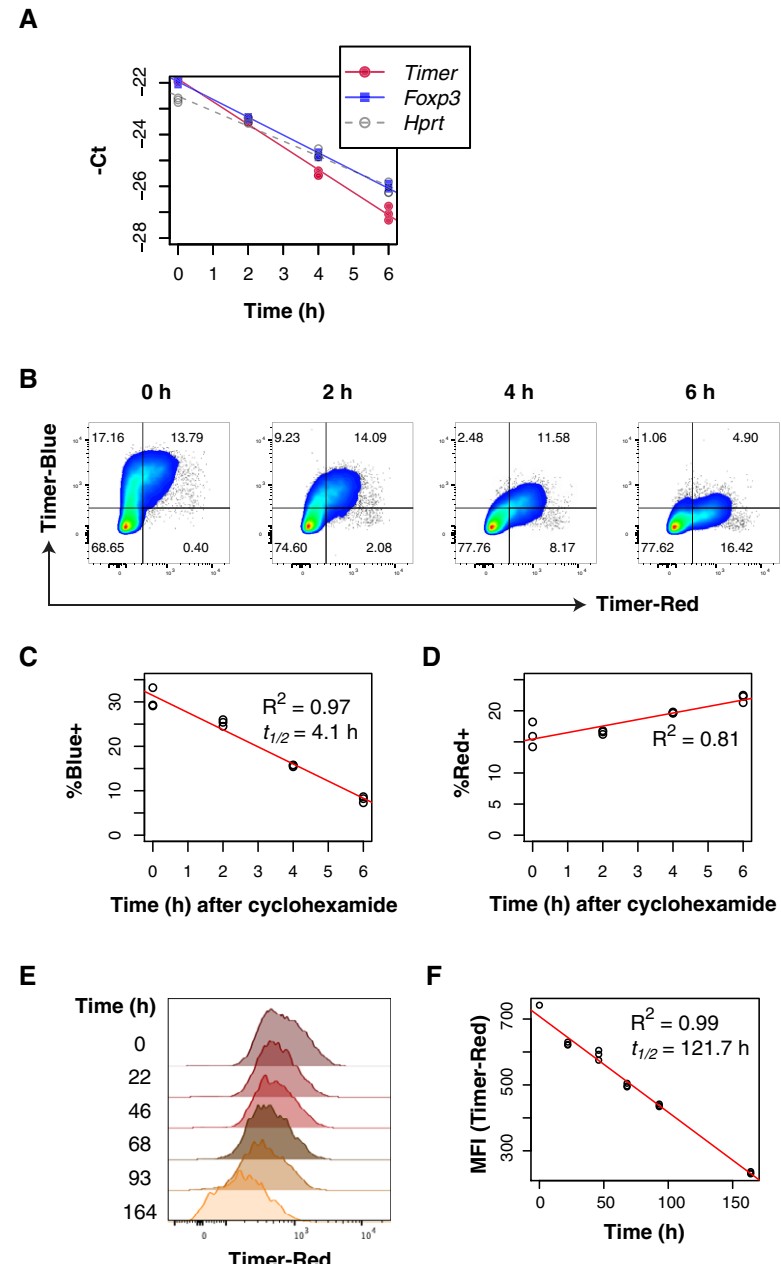

**Figure 1. Timer-Blue fluorescence reports real-time *Foxp3* transcription.**

A    CD4[+] T cells from *Foxp3*-Tocky mice were cultured for the indicated time points with 10 µg/ml actinomycin D. RNA was extracted and *Timer*, *Foxp3* and *Hprt* mRNA detected by RT–PCR. Plotted are the raw Ct values, showing culture triplicates (*n* = 3). Data were reproduced in two independent experiments.

B    Naïve T cells from *Foxp3*-Tocky mice were stimulated for 48 h in the presence of anti-CD3 and anti-CD28 in the presence of IL-2 and TGFβ. After 48 h, cells were harvested and incubated with 100 µg/ml cycloheximide for the indicated time points. Cells were analysed by flow cytometry, and shown is the Timer-Blue versus Timer-Red fluorescence within CD4[+] T cells.

C, D    Summary data of % Timer-Blue[+] or % Timer-Red[+] in the cultures following addition of cycloheximide. Linear regression analysis by Pearson's correlation. Data showing culture triplicates (*n* = 3) were reproduced in two independent experiments.

E    Blue[−]Red[+] CD4[+] T cells from *Nr4a3*-Tocky mice were cultured for the indicated time points and then analysed by histograms for Timer-Red fluorescence in live cells.

F    Linear regression analysis by Pearson's correlation of decay of Timer-Red Mean fluorescence intensity (MFI) in cultures. Data showing culture triplicates (*n* = 3) were reproduced in two independent experiments.

have recently initiated *Foxp3* transcription (Fig 2D). Timer locus analysis showed that splenic Treg remarkably accumulated cells in the PAt and Arrested loci, indicating that the majority of spleen Treg

have less frequent transcription than thymic Treg. Interestingly, the frequency of T cells in the New locus (i.e. T cells that have newly transcribed the *Foxp3* gene in the previous ~4 h) is not much

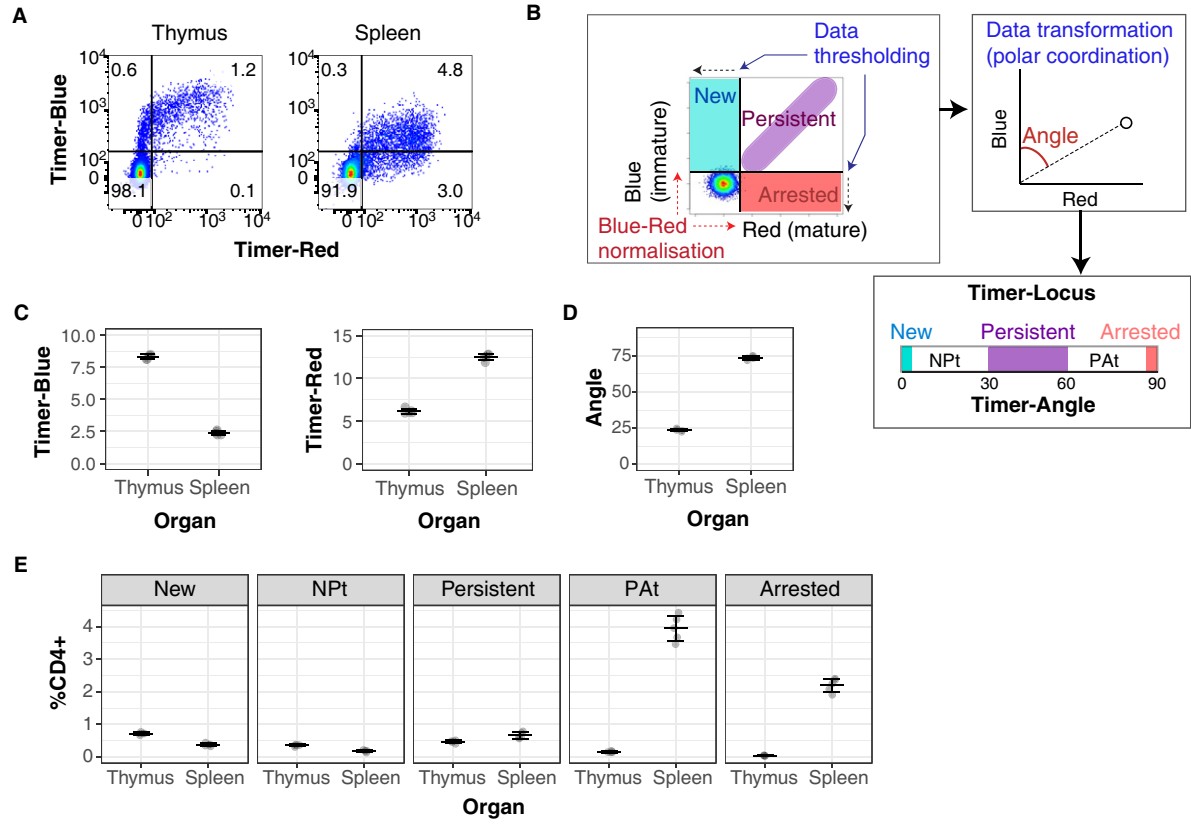

**Figure 2. Thymic CD4-single-positive cells have higher rates of new *Foxp3* transcription compared to splenic CD4[+] T cells in neonatal mice.**

A    CD4-single-positive cells from the thymus and CD4[+] T cells from the spleens of day 10-old *Foxp3*-Tocky mice were analysed for Timer-Blue versus Timer-Red expression by flow cytometry.

B    Timer data analysis schematic (for detailed methodology see Bending *et al*, 2018).

C    Summary data of Timer-Blue and Timer-Red MFI of CD4[+] T cells from thymus or spleen of individual *Foxp3*-Tocky mice, n = 5, error bars represent mean ± SD.

D    Summary data of Timer-Angle in CD4[+] T cells from thymus or spleen of individual *Foxp3*-Tocky mice, n = 5, error bars represent mean ± SD.

E    Timer locus analysis of CD4[+] T cells from thymus or spleen of individual *Foxp3*-Tocky mice, n = 5, error bars represent mean ± SD.

different between the thymus and the spleen from D10 neonates and is ~0.7 and ~0.4% on average, respectively (Fig 2E).

## The flux of new *Foxp3* expressors and the rate of *Foxp3* transcription are increased in tissue-infiltrating T cells during the immune response

Next, we tested whether *Foxp3* transcriptional activity is dynamically regulated during the immune response. We used a skin contact hypersensitivity (CHS) model to analyse the dynamics of *Foxp3* transcription in draining lymph nodes (dLN) and skin-infiltrating T cells after sensitising mice with the hapten oxazolone and challenging them on their ears (Fig 3A). While the effect of hapten on the Timer expression in T cells from dLN showed only marginal changes in the first 5 days during the sensitisation phase (Fig 3B and C), it became more remarkable in the challenge phase. Notably, skin-infiltrating T cells from oxazolone-treated skin had high Timer-Blue expression with lower Timer-Angle values (Fig 3D and E). The distribution (precisely, kernel-density estimation, i.e. a smoothed version of histogram) of Timer-Angle values was markedly right-skewed in dLN, whether vehicle- or oxazolone-treated,

while the Timer-Angle distribution in the skin showed peaks around 30° to 60° (Fig 3E). T cells were not present in sufficient numbers to be recovered from vehicle-treated control skin. Most Timer[+] cells in dLN were Red[+], and these expressed various levels of Blue fluorescence (Fig 3B), which indicates that *Foxp3* transcription levels vary between individual cells. While immunisation with oxazolone resulted in significant increases in *Foxp3* transcription, as evidenced by increase in Timer-Blue and a concomitant reduction in Timer-Angle, in dLN, skin-infiltrating T cells showed uniformly high Timer-Blue fluorescence, indicating that these cells had high-frequency persistent *Foxp3* transcription (Fig 3D and F). On the other hand, Timer-Red fluorescence was lower in skin-infiltrating T cells than dLN cells (Fig 3G), and thus, Timer-Angle was the lowest in skin-infiltrating T cells (Fig 3H). These indicate that these skin T cells had less accumulated *Foxp3* transcripts and thus that these cells are short-lived or rapidly dividing.

Furthermore, we classified cells according to Timer-Angle: New (Angle = 0), Persistent (Angle 30°–60°) and Arrested (Angle = 90°). The areas between New and Persistent and between Persistent and Arrested are designated as NPt and PAt, respectively (Bending *et al*,

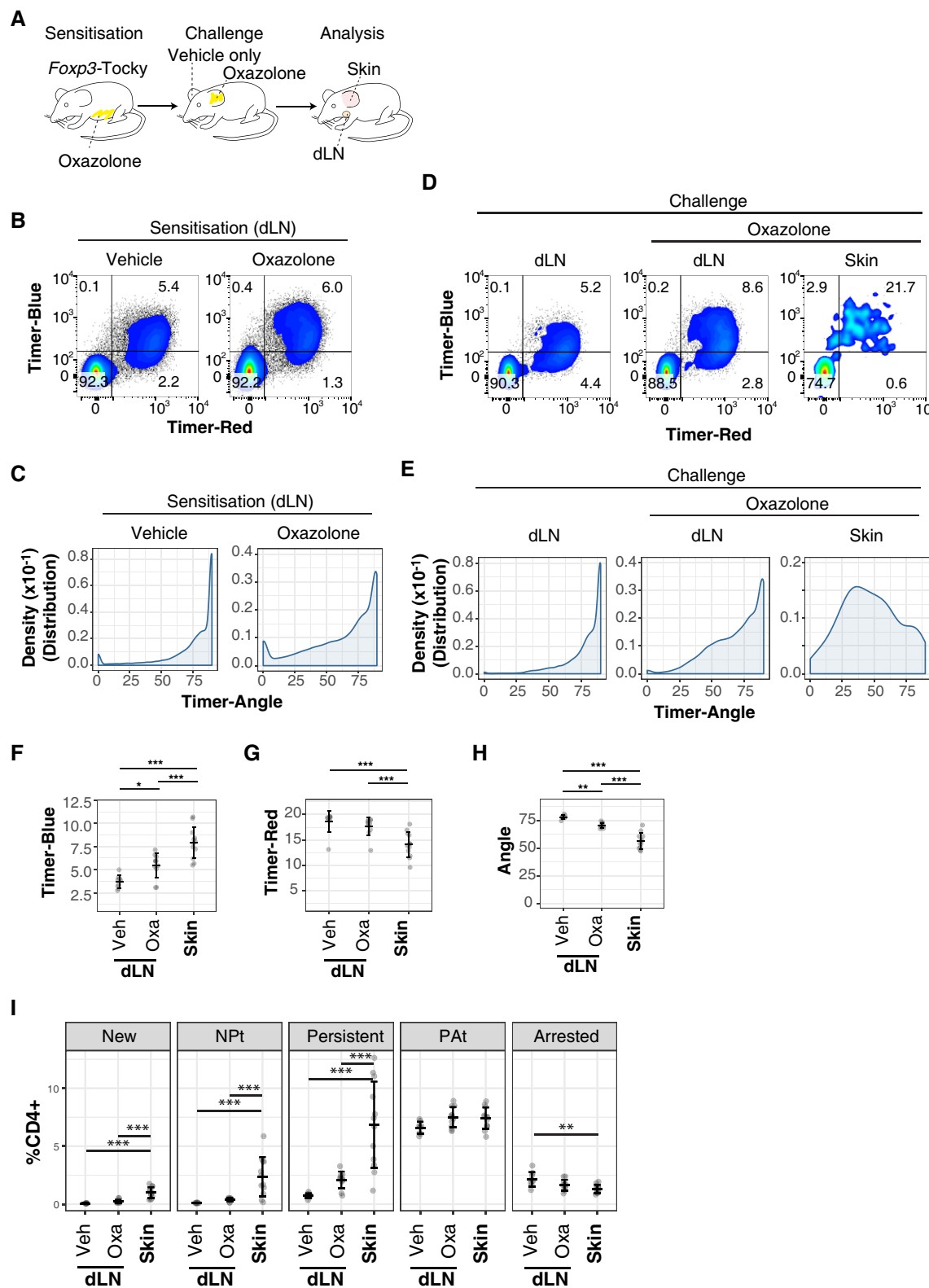

**Figure 3.**

2018; Fig 2B). This Timer locus analysis clearly showed that skin-infiltrating T cells were enriched with New, NPt and Persistent cells, compared to dLN T cells (Fig 3I). Collectively, these results indicate that flux from Foxp3$^-$ to Foxp3$^+$ T cells is increased at sites of inflammation and that skin-infiltrating Foxp3$^+$ T cells have more active *Foxp3* transcription and are short-lived. These cells are

◀

**Figure 3. The flux of new *Foxp3* expressors and the rate of *Foxp3* transcription are increased in tissue-infiltrating T cells during the immune response.**

A   Schematic displaying experimental set-up for analysis of sensitisation and challenge phases of oxazolone-induced CHS using *Foxp3*-Tocky mice.
B   Analysis of Timer-Blue versus Timer-Red fluorescence in CD4[+] T cells from *Foxp3*-Tocky mice 5 days after sensitisation with oxazolone or vehicle control.
C   Kernel-density distributions of Timer-Angle values from (B).
D   48 h after challenge with oxazolone or vehicle control, CD4[+] T cells from *Foxp3*-Tocky mice were analysed. Displayed are Timer-Blue versus Timer-Red flow cytometry plots from vehicle or oxazolone dLN or ear skin.
E   Kernel-density distributions of Timer-Angle values from (D).
F–H   Summary of data showing mean Timer-Blue (F) or Timer-Red intensity (G) or Timer-Angle values (H) in Timer[+] T cells in vehicle dLN ($n = 9$), oxazolone dLN ($n = 11$) or skin ($n = 12$), 24–48 h after ear challenge of mice with oxazolone, error bars represent mean ± SD. *$P < 0.05$, **$P < 0.01$, ***$P < 0.001$. Statistical analysis by one-way ANOVA with Tukey's honest significant difference test.
I   Timer locus analysis of CD4[+] T cells from vehicle dLN ($n = 9$), oxazolone dLN ($n = 11$) or skin ($n = 12$), 24–48 h after ear challenge of mice with oxazolone, error bars represent mean ± SD. **$P < 0.01$, ***$P < 0.001$. Statistical analysis by one-way ANOVA with Tukey's honest significant difference test.

Data information: Data are pooled from four independent experiments.

enriched with newly born cells, which may be undergoing cell division and/or accelerated cell death.

## Foxp3 autoregulation maintains the temporal dynamics of persistent *Foxp3* transcription

The dynamic regulation of *Foxp3* transcription during the immune response led us to hypothesise that Foxp3 protein itself is involved in the regulation of *Foxp3* transcription. In order to understand the mechanism regulating temporally dynamic *Foxp3* transcription, we addressed this possibility by investigating the dynamics of *Foxp3* transcription in the absence of Foxp3 protein. To achieve this, we crossed *Foxp3*-Tocky mice with *Foxp3*[EGFP/Scurfy] mice. Treg from females from this line will all express the Foxp3-driven Timer protein, but half will express Foxp3/EGFP and the other half (EGFP-negative) express the non-functional Scurfy Foxp3 mutant protein (Fig 4A) due to random X-inactivation (Gavin *et al*, 2007; Lin *et al*, 2007). This triple transgenic system thus permits the analysis of *Foxp3* transcriptional activity in the absence of functional Foxp3 protein. *Foxp3*-Tocky *Foxp3*[EGFP/Scurfy] female mice were immunised with oxazolone, and subsequently, *Foxp3* expression within the dLN was analysed. As scurfy T cells are rare, this was performed during the sensitisation phase in order to analyse a larger number of lymph nodes. All Timer[+] T cells were analysed and divided into GFP[+] (WT Foxp3) or GFP[−] (Scurfy Foxp3 mutant) expressing Treg. Interestingly, the majority of Timer[+] cells were Blue[+] in WT Foxp3 cells, while the majority of Timer[+] cells were Blue[−] in Scurfy cells (Fig 4B). Timer-Blue levels were remarkably decreased, and Timer-Red levels were also moderately decreased, in Scurfy cells (Fig 4C). Timer-Angle was significantly higher and approached 90° in Scurfy cells (Fig 4D). Scurfy Timer[+] cells had very few cells displaying Persistent or PAt transcriptional dynamics compared to GFP[+] Treg, with nearly all cells located in the Arrested locus (Fig 4E). Furthermore, in the absence of Foxp3 protein, Timer-expressing Persistent/PAt cells actively transcribed *Il2* (Fig 4F), a characteristic of antigen-stimulated effector T cells. These findings reveal that Foxp3 protein is itself required for controlling *Foxp3* transcription, and the Foxp3 autoregulatory loop actively operates in reactive Treg, suppressing their effector function.

## Persistent *Foxp3* transcription promotes the effector Treg programme

These results suggest that persistent *Foxp3* transcription defines Treg with a unique functional status. In order to address this

hypothesis, we analysed the influence of *Foxp3* transcription status on transcriptional profile using RNA-seq. To our knowledge, currently, no flow cytometer is equipped with the function to use derivative parameters, and it is not possible to sort cells according to Timer-Angle. We sorted cells, therefore, according to their positions in the Blue-Red plane and experimentally confirmed that distinct Timer populations were sorted according to Timer-Angle. T cells were isolated from mice immunised with oxazolone and sorted into four populations by flow cytometry based on their Timer-Blue and Timer-Red expression pattern: "Persistent" *Foxp3* transcriptional expressors were isolated as Pers1 (Blue-intermediate Red-low) and Pers2 (Blue-high Red-high), in addition to PAt (Blue-low Red-high) and Arrested (Blue-negative Red-intermediate) Foxp3 expressors. Sorted cells were analysed by flow cytometry to confirm the purity according to the gate (Fig 5A). Timer analysis showed that each population exhibited discrete Timer-Blue, Timer-Red and Timer-Angle values (Fig 5B–D), confirming the successful flow sorting of the four distinct populations, the mean angles of which corresponded to the desired loci. Principal component analysis (PCA) showed that the persistent *Foxp3* expressors (i.e. Pers1 and Pers2) were clustered and similar to each other, and distinct from PAt and Arrested *Foxp3* expressors, which made another cluster (Fig 5E). Differential expression analysis identified unique clusters of genes (Fig 5F). Persistent *Foxp3* expressors highly expressed the coinhibitory/costimulatory molecules *Ctla4*, *Icos* and *Tigit*, and the serine protease *Gzmb,* all of which are involved in cancer immunity (Sharma *et al*, 2017). In addition, persistent *Foxp3* expressors transcribed other effector Treg signature genes such as *Nkg7*, *Fgl2* and *Irf4*, while they had low expression of naïve T-cell-specific genes such as *Ccr7*, *Bach2* and *Il7r*. These features are fully compatible with the previously reported effector Treg phenotype (Rosenblum *et al*, 2016; Wyss *et al*, 2016). The specific increase in the activation gene *Mki67* (Ki67) in persistent Foxp3 expressors also supports the idea that these cells are highly activated and dividing. Intriguingly, Pearson correlation analysis showed that *Ctla4* and effector Treg markers (*Icos*, *Lag3*, *Tigit*, *Ccr4*) had relatively low correlation to Foxp3 levels ($R^2$ was between 0.30 and 0.65; Fig EV1A), while these markers showed very high correlation to the activation gene, *Mki67* ($R^2$ was between 0.90 and 0.98; Fig EV1B). This suggests that the T-cell activation process predominantly regulates these markers and that persistent Foxp3 expressors express these markers because they are highly activated. In contrast, *Tnfrsf18* (GITR) and *Tnfrsf4* (OX40) showed high correlations to *Foxp3* ($R^2 = 0.91$ and $0.82$, respectively), but only moderate correlations to Mki67 ($R^2 = 0.49$ and $0.74$, respectively), suggesting that Foxp3

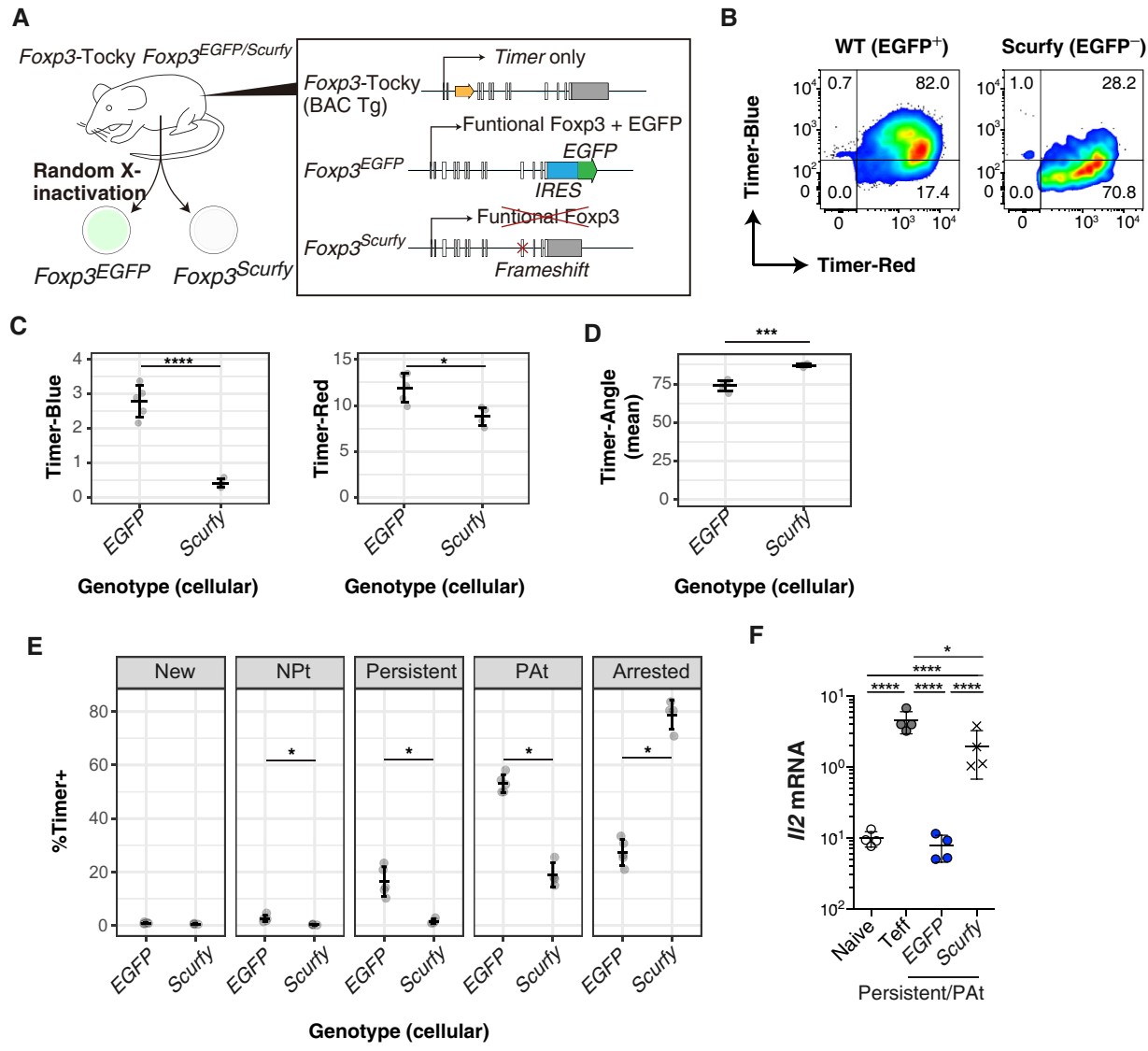

**Figure 4. Foxp3 autoregulation maintains the temporal dynamics of persistent *Foxp3* transcription.**

A   Design of *Foxp3*-Tocky:*Foxp3^EGFP/Scurfy* triple transgenic system to analyse the role of functional Foxp3 in regulation of *Foxp3* transcription. *Foxp3*-Tocky *Foxp3^EGFP/Scurfy* female mice were sensitised with oxazolone for 5 d before analysis of superficial LN (sLN).

B   Shown is Timer-Blue versus Timer-Red expression on Timer[+] T cells within GFP[+] (i.e. WT) or GFP[−] (i.e. scurfy) T cells.[#]

C   Timer-Blue and Timer-Red MFI in Timer[+] T cells from GFP or scurfy expressing Treg. Bars represent mean ± SD, statistical analysis by Student's *t*-test. *$P < 0.05$, ****$P < 0.0001$.

D   Mean Timer-Angle values of *Foxp3^EGFP* and *Foxp3^Scurfy* cells, $n = 4$. Bars represent mean ± SD, statistical analysis by Student's *t*-test. ***$P < 0.001$.

E   Proportion of Timer[+] cells within the five Timer loci in *Foxp3^EGFP* and *Foxp3^Scurfy* from sLN ($n = 4$ mice). Bars represent mean ± SD, statistical analysis by Kruskal–Wallis test. *$P < 0.05$.

F   qPCR for *Il2* mRNA in Naïve, CD44[hi] effector T cells (Teff) and *Foxp3* Persistent cells (Blue[+]Red[+]) from GFP[+] (*Foxp3^EGFP*) and GFP[−] (*Foxp3^Scurfy*). Cells were sorted and RNA was extracted for qPCR, $n = 4$. Bars represent mean ± SD. Data combined from two experiments. Statistical analysis of $\log_{10}$ mRNA expression values by one-way ANOVA with Tukey's honest significant difference test. *$P < 0.05$, ****$P < 0.0001$.

may more closely regulate GITR and OX40. *Il2ra* is ubiquitously highly expressed by all Foxp3[+] cells, and not by Foxp3[−] cells, and showed only low correlations to both *Foxp3* and *Mki67* levels ($R^2 = 0.26$ and $0.43$), suggesting that CD25 expression is stabilised in Foxp3[+] cells but regulated primarily by other factors, such as IL-2 signalling once Foxp3 is expressed (Fig EV1A and B). Persistent

Foxp3 expressor-specific genes were enriched with cell cycle-related pathways. In contrast, the PAt and Arrested-specific genes were enriched with cytokine signalling pathways but not cell cycle-related pathways (Fig 5G). Collectively, RNA-seq analyses showed that temporally persistent *Foxp3* transcription is an underlying mechanism for effector Treg differentiation.

---

[#]Correction added on 20 July 2018, after first online publication: the labels WT (EGFP[+]) and Scurfy (EGFP[−]) were swapped in Fig 4B.

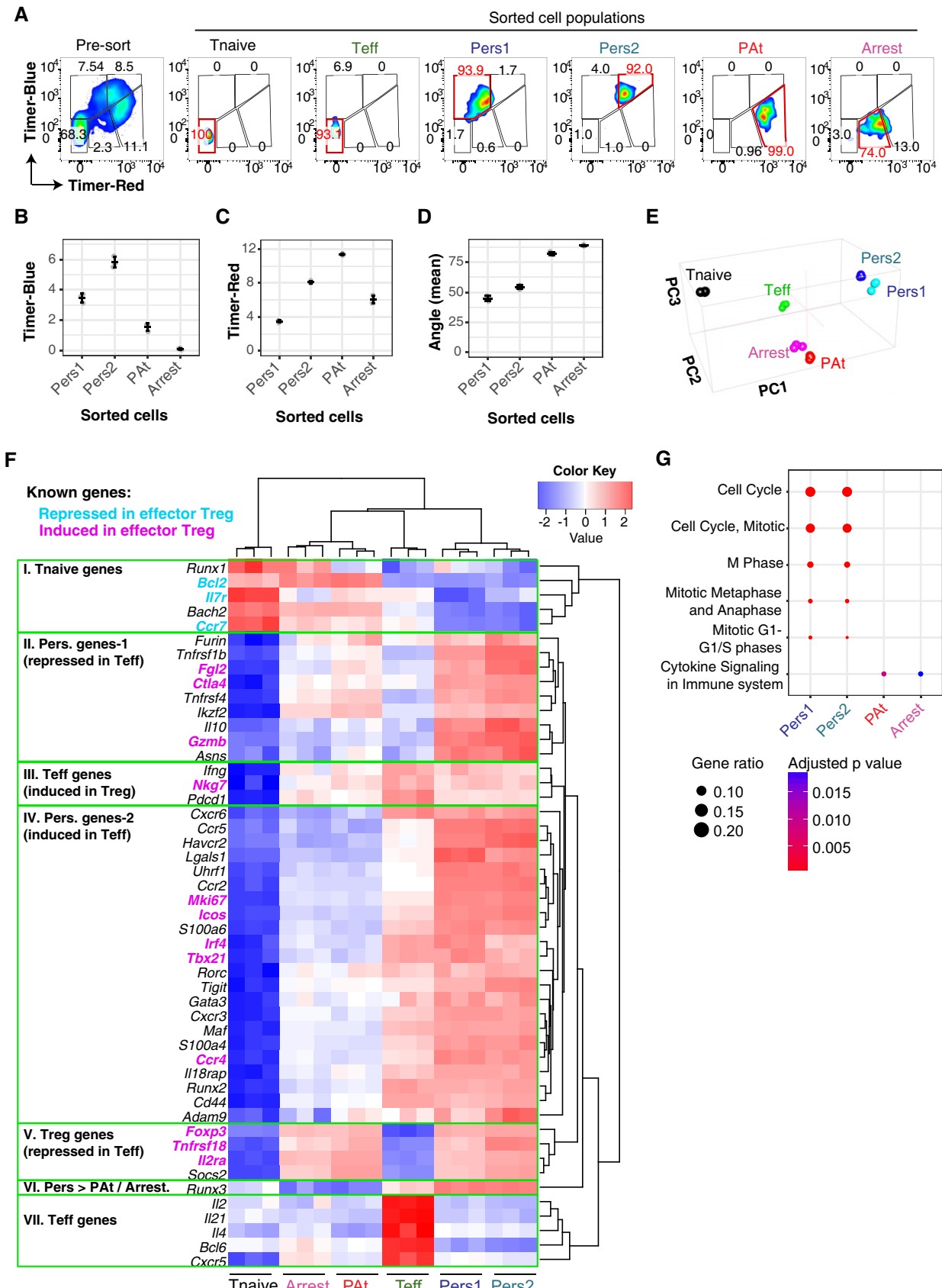

**Figure 5.**

**Figure 5. Persistent *Foxp3* transcription promotes the effector Treg programme.**

A    *Foxp3*-Tocky mice were sensitised with oxazolone for 5 days and dLN harvested and stained with CD4 and CD44 for sorting. Pre- and post-sort purities displaying Timer-Blue versus Timer-Red fluorescence for Tnaive (CD4⁺CD44ˡᵒTimer⁻) and Teff (CD4⁺CD44ʰⁱTimer⁻). Timer⁺ T cells from sensitised *Foxp3*-Tocky mice were sorted into Pers1, Pers2 PAt and Arrested T cell subsets, according to the maturation of Timer chromophore. A small proportion of sorted cells were analysed by flow cytometry. Red gate indicates the one used for sorting each population, and red number indicates the purity in terms of Timer fluorescence. $n$ = 3 groups, each made from pools of three mice.

B–D    Mean Timer-Blue, Timer-Red fluorescence intensity levels or Timer-Angle in the sorted Timer⁺ subsets, $n$ = 3. Bars represent mean ± SD.

E    RNA was extracted from the samples, and sequencing was performed as described in the methods. Cluster analysis of the transcriptomes of the sorted samples by PCA.

F    Heatmap analysis of differentially expressed genes (DEGs) in the four Timer⁺ subsets, compared to Tnaive and Teff, grouping genes into seven categories.

G    Pathway analysis of the four DEG gene lists.

## Identification of distinct groupings of cell surface markers for targeting T cells with specific *Foxp3* transcriptional dynamics

Next, we analysed protein expression of candidate surface proteins identified by RNA-seq for targeting T cells with specific *Foxp3* transcriptional dynamics *in vivo*. T cells from *Foxp3*-Tocky mice were immunised with oxazolone and analysed by flow cytometry for the expression levels of candidate markers from Fig 5 in each Timer locus. Hierarchical clustering of MFI levels showed that these candidate genes were grouped into two main groups, in addition to Neuropilin 1 (Nrp1) which had its own separate cluster (Fig 6A). Group I markers showed high expression throughout New–Persistent *Foxp3* expressors, but fell drastically as Blue fluorescence decreases in PAt and Arrested *Foxp3* expressors (Fig 6B), suggesting that these markers are associated with the initiation of new *Foxp3* transcription in activated T cells and with high-frequency *Foxp3* transcription. Group II markers are less expressed in new *Foxp3* expressors, and increased in persistent and PAt *Foxp3* expressors (Fig 6C), suggesting that these markers can be sustained by moderate frequencies of *Foxp3* transcription once the Foxp3 autoregulatory loop is established. Nrp1 is characteristically high in PAt *Foxp3* expressors, which may explain why this marker has been used to identify thymus-derived Treg (Fig 6D). Importantly, the expression levels of both Group I and II markers markedly declined in Arrested *Foxp3* expressors. Together, these results further support the notion that the temporal frequency (rate) of *Foxp3* transcription regulates the Treg phenotype and function.

## *Foxp3*-Tocky allows visualisation of the manipulation of T cells with specific *Foxp3* transcriptional dynamics upon immunotherapy

We asked whether surface proteins from Group I and Group II in Fig 6 can be used to target T cells with specific *Foxp3* transcriptional dynamics. If successful, this approach may lead to future precision immunotherapy for controlling Foxp3-mediated immune regulation by manipulating its temporal dynamics. Both TNFRII (Group I) and OX40 (Group II) are targets for immunotherapy and may be used to manipulate T-cell responses, and currently, clinical trials are ongoing for melanoma and other solid cancers (Croft *et al*, 2009). However, it is unknown whether these immunotherapy antibodies affect *Foxp3* transcriptional dynamics.

Using the CHS model, antibodies were administered during the challenge phase (Fig 7A). Treatment with anti-TNFRII antibody increased the frequency of Blue⁺Red⁻ T cells and Blue⁺Red⁺ T cells at the inflamed site (Fig 7B). Although average Blue and Red levels were overall not affected by the antibody (Fig 7C and D), Timer-Angle tended to be lower within the anti-TNFRII-treated group ($P$ = 0.055), likely reflecting the increased proportion of pure blue T cells at the inflamed site (Fig 7E). Analysis of the Timer-Angle distribution revealed that isotype control treatment showed one dominant peak around the 45° mark, while anti-TNFRII treatment produced a more flattened, peak from 25 to 50 degrees (Fig 7F). Analysis of Timer locus revealed that anti-TNFRII increased both New and NPt stage *Foxp3* expressors, indicating that anti-TNFRII immunotherapy may enhance the flux of Foxp3⁻ into Foxp3⁺ T cells (Fig 7G). There was no significant change in the high-frequency/persistent *Foxp3* transcribers, and anti-TNFRII treatment did not significantly alter the course of inflammation (Fig 7H). In contrast, anti-OX40 treatment reduced the frequency of Blue⁺Red⁺ Foxp3⁺ T cells in inflamed skin (Fig 7I). While Blue levels were unaffected (Fig 7J), both Timer-Red and Timer-Angle were significantly lower in anti-OX40-treated mice (Fig 7K and L). Interestingly, the peak of Timer-Angle distribution in anti-OX40-treated mice was shifted towards the NPt (Fig 7M). Timer locus analysis revealed that anti-OX40 reduced the frequency of more mature *Foxp3* expressors, as both Persistent and PAt loci were significantly reduced in anti-OX40-treated mice (Fig 7N). These data indicate that, in the presence of anti-OX40 antibody, Treg are very short-lived at the inflamed site and that a proportion of high-frequency *Foxp3* expressors are either depleted or undergo activation-induced death. In fact, a majority of skin-infiltrating OX40⁺ T cells were stained by Annexin V while OX40⁻ counterparts were not (Fig EV2). Interestingly, anti-OX40 treatment delayed the resolution of inflammation (Fig 7O). These data together suggest that effector Treg in the skin are pre-apoptotic-activated cells that function as suppressor cells and that anti-OX40 antibody accelerated the death of pre-apoptotic-activated Foxp3⁺ T cells by the mechanism of activation-induced cell death. In summary, both of the effects of the immunotherapies tested were in keeping with the *Foxp3* transcriptional dynamics identified in Fig 6, showing how *Foxp3*-Tocky may be used to identify new therapeutic targets and mechanism of drug action in preclinical models.

## Discussion

Sustained *Foxp3* transcription is a cardinal feature of the functional maturation of Treg (i.e. effector Treg) (Fig 5). Importantly, sustained *Foxp3* transcription requires functional Foxp3 protein itself. This type of transcriptional regulation (i.e. transcription factor regulates its own gene) is defined as positive autoregulation, which allows a

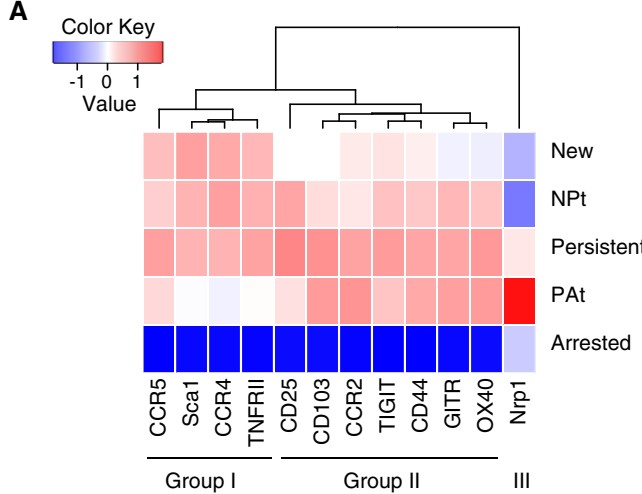

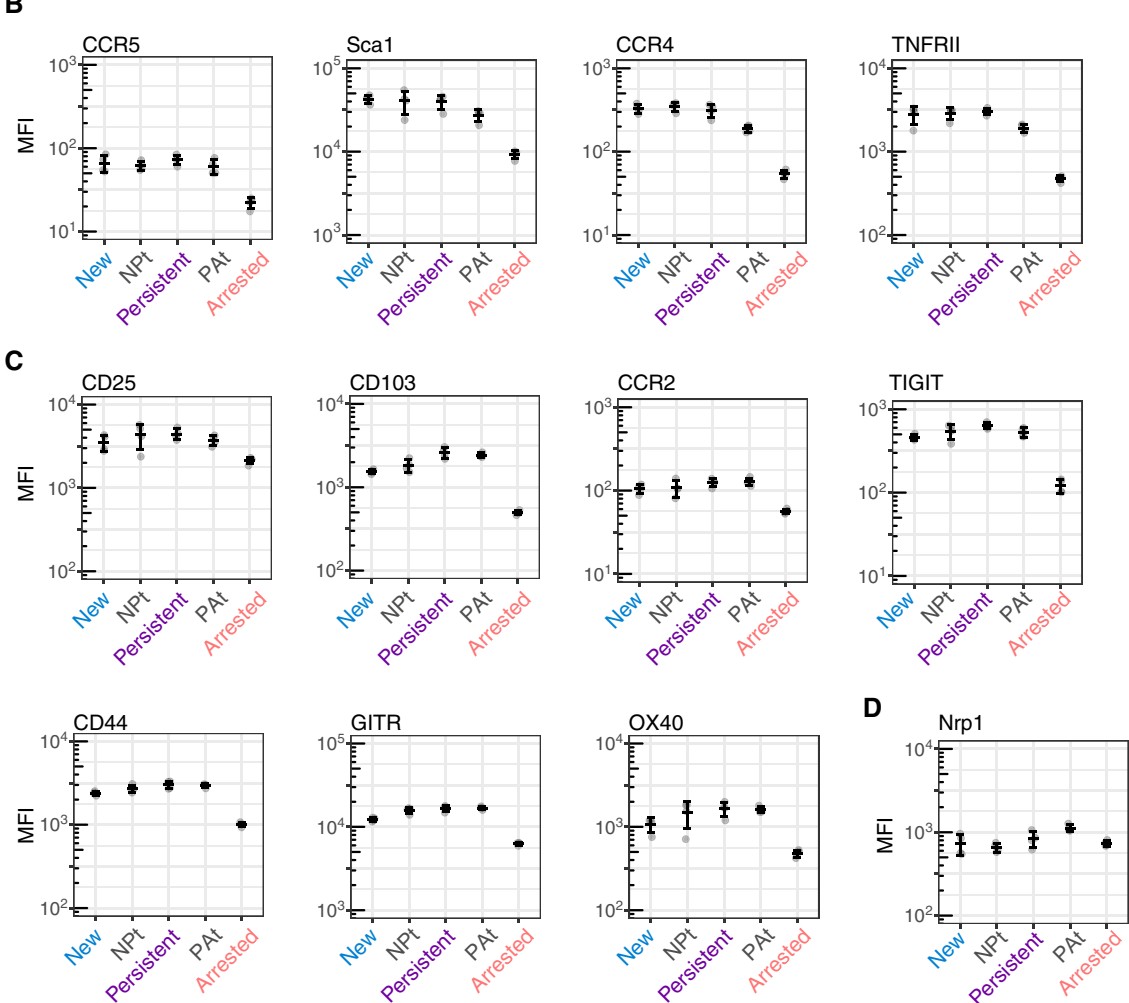

**Figure 6.  Identification of distinct groupings of cell surface markers for targeting T cells with specific *Foxp3* transcriptional dynamics.**

A     *Foxp3*-Tocky mice were sensitised with oxazolone for 4 days, and then, dLN were harvested and CD4$^+$ T cells analysed for Timer expression and key membrane proteins identified as differentially expressed by RNA-seq. Heatmap analysis of the expression of different surface proteins according to cells within different Timer loci. These findings generate three main groups.

B–D   MFI expression levels of Group I, Group II or Group III surface markers in relation to Timer loci of the cells, *n* = 4. Bars represent mean ± SD.

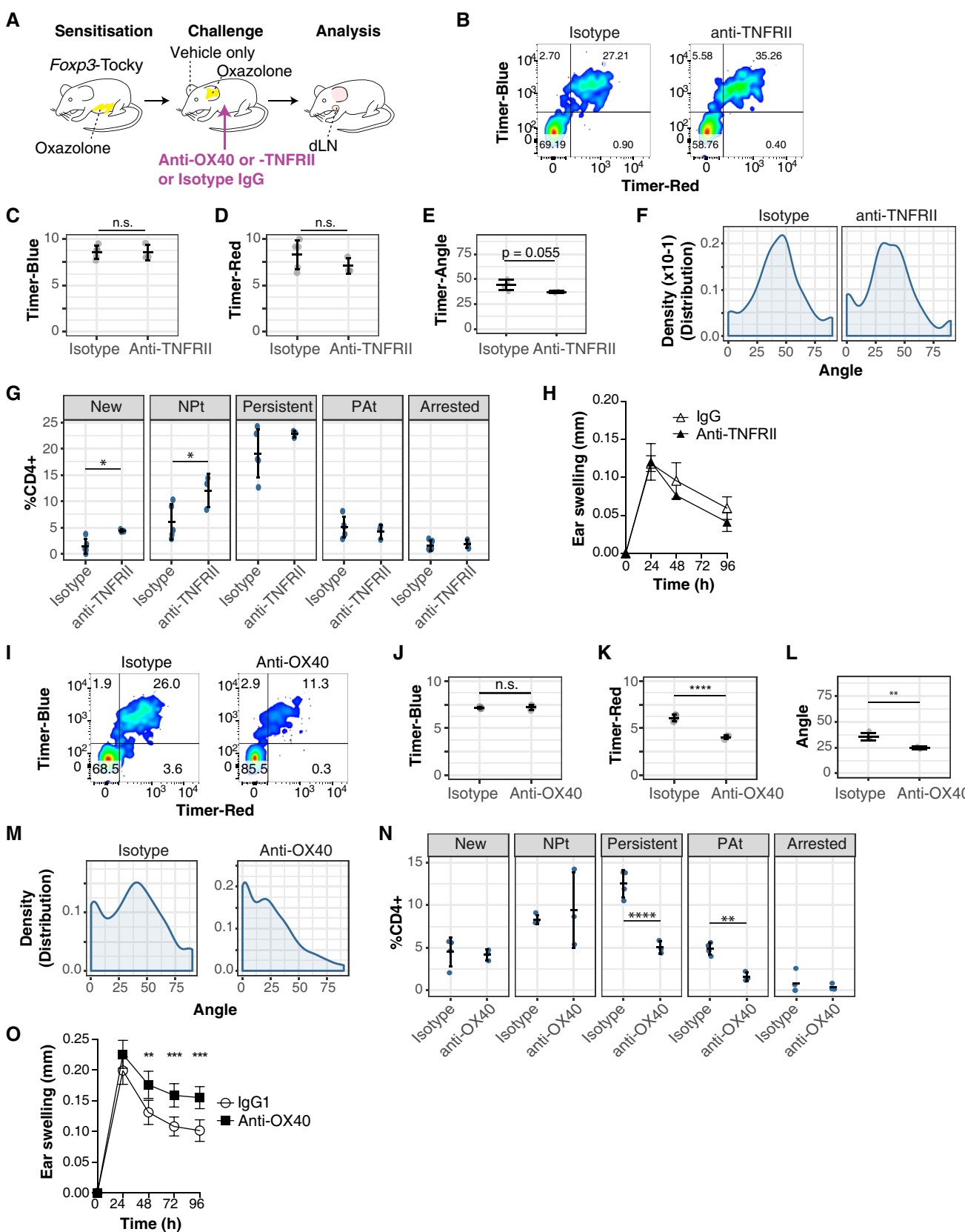

**Figure 7.**

**Figure 7.   *Foxp3*-Tocky allows visualisation of the manipulation of T cells with specific *Foxp3* transcriptional dynamics upon immunotherapy.**

A       Experimental design for the administration of anti-OX40 or anti-TNFRII antibody to target the challenge phase of CHS.

B       Mice were challenged with oxazolone and then administered isotype or anti-TNFRII antibodies. 96 h later, CD4[+] T cells from the skin were analysed for expression of Timer-Blue versus Timer-Red (concatenated plots are shown).

C–E    Summary of Timer-Blue MFI, Timer-Red MFI or Timer-Angle in skin-infiltrating Timer[+] T cells, isotype $n = 5$, anti-TNFRII $n = 3$. Bars represent mean $\pm$ SD.

F       Kernel-density distributions of Timer[+] T cells in isotype or anti-TNFRII treated mice 96 h after challenge with oxazolone.

G       Timer locus distribution of skin-infiltrating CD4[+] T cells in isotype or anti-TNFRII-treated *Foxp3*-Tocky mice. Bars represent mean $\pm$ SD. Statistical analysis by one-way ANOVA with Tukey's honest significant difference method. *$P < 0.05$

H       Ear swelling during the challenge phase of CHS in isotype or anti-TNFRII treated *Foxp3*-Tocky mice, bars represent mean $\pm$ SEM.

I       Mice were challenged with oxazolone and then administered isotype or anti-OX40 antibodies. 96 h later, CD4[+] T cells from the skin were analysed for expression of Timer-Blue versus Timer-Red (concatenated plots are shown).

J–L    Summary of Timer-Blue MFI, Timer-Red MFI or Timer-Angle in skin-infiltrating Timer[+] T cells, isotype $n = 4$, anti-OX40 $n = 3$. Bars represent mean $\pm$ SD. Statistical analysis by Student's *t*-test. **$P < 0.01$, ****$P < 0.0001$.

M       Kernel-density distributions of Timer[+] T cells in isotype or anti-OX40 treated mice 96 h after challenge with oxazolone.

N       Timer locus distribution of skin-infiltrating CD4[+] T cells in isotype ($n = 4$) or anti-OX40 ($n = 3$)-treated *Foxp3*-Tocky mice. Bars represent mean $\pm$ SD. Statistical analysis by one-way ANOVA with Tukey's honest significant difference test. **$P < 0.01$, ****$P < 0.0001$.

O       Ear swelling during the challenge phase of CHS in isotype ($n = 19$) or anti-OX40 ($n = 18$)-treated mice. Data are combined from two experiments and show mean $\pm$ 95% confidence intervals. Statistical analysis of 24-, 48-, 72- and 96-h time points by two-way ANOVA with Sidak's multiple comparisons test. **$P < 0.01$, ***$P < 0.001$.

slow and sustained transcriptional response to external signalling cues (Yosef & Regev, 2011). This may constitute the mechanism behind effector Treg differentiation in inflammatory environments. It is known that, among the conserved non-coding DNA sequence (CNS) elements of the *Foxp3* gene, the CNS2, which contains the TSDR, is required for maintenance of Foxp3 expression after cell division (Zheng *et al*, 2010). The CNS2 regions are bound by Foxp3 (Zheng *et al*, 2010), Runx1/Cbf-β (Kitoh *et al*, 2009; Rudra *et al*, 2009) and Stat5 (Yao *et al*, 2007). Intriguingly, the genetic deletion of CNS2 or Cbf-β results in the reduction in Foxp3 expression in Treg, especially in activated Treg, and the development of autoimmune inflammation (Kitoh *et al*, 2009; Feng *et al*, 2014). These suggest that the autoregulatory circuit of the *Foxp3* gene is mediated through the Foxp3-Runx1 interaction (Ono *et al*, 2007) at the CNS2 (Rudra *et al*, 2009), and persistent *Foxp3* transcription in activated Treg is essential for establishing the effector Treg programme and maintaining immunological tolerance. Given that the autoregulatory transcriptional circuit sustains *Foxp3* transcription, genes with linear correlations to *Foxp3* transcripts (e.g. *Tnfrsf4* and *Tnfrsf18*, Fig EV1) are likely regulated by this transcriptional circuit. On the other hand, genes with high expression in persistent *Foxp3* expressors that show moderate correlations to *Foxp3* levels (e.g. *Ctla4*, Fig EV1A) are likely to be regulated by both the T-cell activation process and Foxp3 autoregulation-dependent mechanisms. Thus, in activated effector Treg, the autoregulatory loop of the *Foxp3* gene and the T-cell activation process may cooperatively control transcellular suppressive mechanisms (e.g. CTLA-4), while suppressing the effector functions (e.g. cytokine production) in a cell-intrinsic manner. A future study may focus on the identification of molecular mechanisms for the establishment and control of the *Foxp3* autoregulatory transcriptional circuit across chromatin regions and/or at the single cell level.

Interestingly, the induction of new *Foxp3* expression may be more frequent in physiological conditions than previously thought. As the half-life of Blue fluorescence is very short, in young mice, peripherally induced Foxp3 expression may occur in significant part of CD4[+] T cells (up to 0.3% per 4 h, based on Day 10 New cells in spleen), and these Foxp3[+] cells can accumulate rapidly unless they die or terminate Foxp3 expression. In the

hapten model, ~10% of Timer[+] cells are identified at the New locus in the skin (Fig 3I). Given that these new Foxp3 expressors quickly move to NPt/Persistent locus within 4 h (Fig 1B), a significant proportion of cells at the Persistent locus may have been supplied from these newly generated Treg before the analysis. It is also likely that cells at the PAt and Arrested loci move to the Persistent locus by increasing *Foxp3* transcription. Since 1–5% of Foxp3-CD4[+] T cells acquire new Foxp3 expression per 4 h (Figs 1C, 3I, and 7G and N), these new Foxp3 expressors may be short-lived. These two possibilities are not mutually exclusive and are compatible with evidence that the effector Treg population arises from both thymic and peripherally induced Treg (Rosenblum *et al*, 2016), while it is predicted that the ratio of the contributions may vary between antigens and between individuals (Ono & Tanaka, 2016).

Intriguingly, the majority of mature Foxp3 expressors in the inflamed skin are OX40[high] and Annexin V[+]. While Annexin V is often considered as a marker of apoptotic cells, Annexin V[+] activated T cells are still alive and can be functional to preclude the development of T-cell memory (Wang *et al*, 2004). Effector Treg in the inflamed tissue may be in fact such functional pre-apoptotic cells. Thus, our study supports the model that tissue-infiltrating effector Treg are actively generated from activated T cells during inflammation through the autoregulatory transcriptional circuit of the *Foxp3* gene. Most probably, due to their sustained *Foxp3* expression, these cells may function as pre-apoptotic decoy cells that consume cytokines and thereby suppress the activities of other T cells until they die. This model is also compatible with the pro-apoptotic role of Foxp3 under cytokine deprivation (Tai *et al*, 2013), which may be important for shrinking the expanded Treg population during the resolution phase of inflammation. These context-dependent mechanisms of Foxp3 function in T cells may underlie the pleiotropic mechanisms of anti-OX40: OX40 costimulation enhances the survival of Foxp3[−] effector T cells (Croft *et al*, 2009), while it may rather accelerate the death of Foxp3[+] effector Treg.

It may be important to emphasise that the proliferated Treg will die after resolution of inflammation, as other T cells do (McKinstry *et al*, 2010). If some of the reactive Foxp3 expressors

survive beyond the resolution of inflammation, such cells will be memory T cells by definition. Some cells may sustain Foxp3 expression and survive as "memory Treg", and some others may lose Foxp3 expression after the resolution of inflammation and survive as memory-phenotype (memory-like) T cells (Ono & Tanaka, 2016). Intriguingly, ex-Treg are enriched in the memory-phenotype T-cell pool and prone to express Foxp3 more readily than naïve T cells (Zhou *et al*, 2009; Miyao *et al*, 2012). Since, in non-inflammatory conditions, a majority of Foxp3$^+$ T cells are at the PAt or Arrested loci (Figs 2A and 3B), and these cells are similar to each other and distinguished from functional effector Treg at the transcriptome level (Fig 5E), all these non-reactive Treg cells may be non-functional and constitute a spectrum of "memory" T cells against self-antigen, together with naturally arising memory-phenotype T cells. Importantly, both Foxp3$^+$ Treg and memory-phenotype T cells infrequently and spontaneously receive "tonic" TCR signals during homeostasis (Bending *et al*, 2018), which are required for maintaining the size of Treg population and Foxp3 expression (Feng *et al*, 2014; Vahl *et al*, 2014). These together support the model that self-reactive T cells may have fluctuating Foxp3 expression and can be identified as either Foxp3$^+$ Treg (PAt or Arrested) or memory-phenotype T cells at different time points. These hypotheses will be best addressed in future studies using new approaches to combine the Tocky technology and single cell technologies.

The current study has shown that *Foxp3*-Tocky is effective in designing immunotherapy strategies to target a specific phase of *Foxp3* transcriptional dynamics, and also, for visualising alterations of the dynamics upon immunotherapy. In fact, we successfully manipulated *in vivo* Foxp3 dynamics by targeting Timer stage-specific surface receptors. By targeting Group I (TNFRII) or Group II (OX40) marker of *Foxp3* dynamics (Fig 6), we either increased the flux of new *Foxp3* transcription (anti-TNFRII) or decreased mature *Foxp3* expressors (anti-OX40). While the precise cellular mechanism is to be determined, anti-OX40 treatment resulted in the loss of persistent and PAt *Foxp3* expressors within the skin, which are enriched with effector Treg. OX40 is an emerging target for cancer immunotherapy, and its agonistic antibodies for augmenting the effector function of T cells are currently tested in clinical trials (Aspeslagh *et al*, 2016). It is currently thought that OX40 is expressed mainly by activated effector T cells and is also constitutively expressed on all Treg (Weinberg *et al*, 2011). Such OX40 high expression is found relatively more frequently on persistent Foxp3 expressors compared to Foxp3-negative effector T cells.

In summary, we have dissected the previously hidden dynamics of *Foxp3* transcription that regulate the functions of Treg through time. In addition, we have shed light on how Foxp3-mediated molecular mechanisms coordinate T-cell responses in existing Treg and non-Treg, introducing a new dimension into studies on Foxp3-driven T-cell regulation.

# Materials and Methods

## Study design

Sample size for anti-OX40 CHS experiments was estimated based on a power calculation with alpha = 0.05 and power 80%, which gave a sample size of 8. For analysis of *Foxp3*-Tocky *Foxp3$^{EGFP/Scurfy}$* mice, four mice were analysed over two independent experiments (age 6–10 weeks). For anti-OX40 experiments, a pilot experiment was performed to obtain estimates for effect size and standard deviation, which yielded an experimental group size of 9. Two independent experiments were then performed to test the null hypothesis that anti-OX40 treatment had no effect on ear swelling at 96 h. The null hypothesis was rejected in both experiments, with no significant inter-experiment variation. Therefore, data from both experiments were pooled for presentation of results. For all other experiments, a minimum of three animals was used. Where male and female mice were used in the same study, these were randomised into experimental groups according to sex and age. Similarly, wild-type littermates from transgenic mice were randomised into experimental groups for the anti-OX40 experiments. The individual making the ear measurement was blinded to the treatment group of the mouse until after the measurement was taken.

## Transgenesis and mice

*Foxp3*-Tocky and *Foxp3*-Tocky:*Foxp3-IRES-GFP* transgenic reporter strains were generated as described (Bending *et al*, 2018). The triple transgenic *Foxp3*-Tocky *Foxp3$^{EGFP/Scurfy}$* mice were generated by crossing *Foxp3*-Tocky mice with *Foxp3$^{tm9(EGFP/cre/ERT2)Ayr}$*/J and B6.Cg-*Foxp3$^{sf}$*/J (Jackson Laboratories, #016961 and #004088, respectively). All animal experiments were performed in accordance with local Animal Welfare and Ethical Review Body at Imperial College London (Imperial) and University College London (UCL), and all gene recombination experiments were performed under the risk assessment that was approved by the review board at Imperial and UCL.

## *In vitro* cultures for determining mRNA half-lives

CD4$^+$ T cells from the spleens of *Foxp3*-Tocky mice were isolated by immunomagnetic separation (StemCell Technologies) and cultured at $4 \times 10^5$ cells per well in the presence of 10 µg/ml actinomycin D. Cells were isolated at the indicated time points, and RNA was extracted, and cDNA synthesises as described below.

## *In vitro* cultures for determining Timer-Blue and Timer-Red half-life

Naïve T cells from *Foxp3*-Tocky mice were isolated by negative selection using immunomagnetic selection (StemCell Technologies) and $2 \times 10^5$ cells cultured on anti-CD3 (clone 1452C11, 2 µg/ml) and anti-CD28 (clone 37.51, 10 µg/ml; both eBioscience)-coated 96-well plates (Corning) in the presence of 500 U/ml rhIL-2 (Roche) and 5 ng/ml rhTGFβ (R&D) for 48 h in a final volume of 200 µl RPMI1640 (Sigma) containing 10% FCS and penicillin/streptomycin (Life Technologies). Cells were harvested then incubated with 100 µg/ml cycloheximide (Sigma) for the indicated time points before analysis by flow cytometry. For determination of the red half-life, Blue$^-$Red$^+$ CD4$^+$ T cells from *Nr4a3*-Tocky mice were isolated by cell sorting and then cultured without TCR signals under the presence of rhIL-2 for the indicated number of hours before analysis by flow cytometry. Importantly, IL-2 signals do not induce Timer expression in *Nr4a3*-Tocky T cells (Bending *et al*, 2018).

## Contact hypersensitivity and antibody treatments

Oxazolone (Sigma) solutions were prepared fresh in ethanol for each experiment. For sensitisation, 150 µl of 3% oxazolone was applied to the shaven abdominal skin of anaesthetised 5- to 10-week-old mice. Five days later, left and right ear thickness were measured, and the elicitation phase started by the application of 20 µl of a 1% oxazolone solution to one ear, or 20 µl of vehicle (100% ethanol) to the contralateral ear. Twenty-four to 96 h later, ear thickness was measured using a digital micrometer (Mitutoyo). In anti-OX40 experiments, 0.5 mg of anti-OX40 (OX86, BioXCell) or rat IgG1 (MAC221, kind gift from Prof. Anne Cooke) was injected i.p. on day of challenge and 48 h later. For anti-TNFRII experiments, 0.5 mg of anti-TNFRII (TR75-54.7 BioXCell) or Armenian hamster IgG (BioXCell) was injected intra-peritoneally (i.p). on day of challenge.

## Flow cytometric analysis and cell sorting

Following spleen or thymus removal, organs were forced through a 70-µm cell strainer to generate a single cell suspension. For splenocyte preparations, a RBC-lysis stage was employed. Ears were removed and cut into small fragments, which were forced through a 70-µm cell strainer, to make a single cell suspension. Cells were washed once before re-filtering through a 50-µm cell strainer. Staining was performed on a V-bottom 96-well plate or in 15-ml falcon tubes for cell sorting. Analysis was performed on a BD FACS Aria III (Figs 2–5 and 7) or Fortessa (Figs 1, 6 and EV2) instrument. The blue form of the Timer protein was detected in the blue (450/40 nm) channel excited off the 405 nm laser. The red form of Timer protein was detected in the mCherry (610/20) channel excited off the 561-nm laser. For all experiments, a fixable eFluor 780-fluorescent viability dye was used (eBioscience). The following directly conjugated antibodies were used in these experiments: CD4 APC (clone RM4-5, eBioscience), CD4 Alexa Fluor 700, CD4 BUV395 (clone GK1.5 BD Biosciences), (clone RM4-5, BioLegend), CD8 PE-Cy7 (clone 53-6.7, BioLegend), CD8 BUV737 (clone 53-6.7, BD Biosciences), TCRβ FITC & Alexa Fluor 700 (clone H57-597, BioLegend), TCRβ BUV737 (clone H57-597, BD Biosciences), CD25 PerCPcy5.5 (PC61.5, eBioscience) or PE-Cy7 (PC61.5, Tonbo Bioscience), CD44 APC (clone IM7, eBioscience) or Alexa Fluor 700 (clone IM7, BioLegend), OX40 PE-Cy7 and APC (clone OX86, BioLegend), GITR FITC (clone DTA-1, BioLegend), TNFRII APC (clone TR7554, R&D Systems), Neuropilin-1 APC or PE-Cy7 (clone 3E12 BioLegend), CD103 APC (clone 2E7, BioLegend), CCR4 APC (clone 2G12, BioLegend), TIGIT PE-Cy7 (Clone 1G9, BioLegend), CCR2 AF647 (clone SA203G11, BioLegend), CCR5 APC (clone HM-CCR5, BioLegend) and Sca1 (clone D7, BioLegend). Annexin V staining was performed using APC-labelled Annexin V, according to the manufacturer's instructions (eBioscience).

## qPCR analysis

For sorted cells, RNA was extracted using the Arcturus PicoPure RNA kit (Life Technologies) according to the manufacturer's instructions. cDNA was generated using random hexamers and Superscript II (Life Technologies) according to the manufacturer's instructions. mRNA expression was quantified using SYBR green (BioRad) and

expressed relative to the housekeeping gene *Hprt*. Primer sequences: *Hprt* For: AGCCTAAGATGAGCGCAAGT, *Hprt* Rev: TTACTAGGCA GATGGCCACA. *Il2* For: AGCAGCTGTTGATGGACCTA *Il2* Rev: CGCAGAGGTCCAAGTTCAT (Martins *et al*, 2008). *Foxp3* For: CACCCAAGGGCTCAGAACTTCTAG, *Foxp3* Rev: ATGACTAGGGG CACTGTAGGCA, *Foxp3-Timer* For: CAGCTCCTCTGCCGTTATCC, *Foxp3-Timer* Rev: CCTCGCCCTCGATCTCGA

## RNA-seq

*Foxp3*-Tocky mice were sensitised by application of 3% oxazolone to the abdomen. Five days later, dLN were harvested and CD44$^{lo}$Foxp3$^-$, CD44$^{hi}$Foxp3$^-$, *Pers1, Pers2, PAt* and *Arrested* T cells were sorted and RNA extracted using the Arcturus PicoPure RNA kit (Life Technologies) according to the manufacturer's instructions. Library was prepared using the Illumina TruSeq® Stranded mRNA LT kit according to manufacturer's instructions. Libraries were analysed by Bioanalyzer (Agilent) for quality control and quantified using Qubit (Life Technologies). Library concentrations were normalised and pooled and sequenced on the Illumina HiSeq platform (Illumina), and 100-bp paired-end readings were obtained. After demultiplexing fastq files and performing QC analysis by *fastQC*, sequence reads were aligned to either the mouse genome (mm10) with or without the *Timer* gene by *TopHat2*. Statistical analysis was performed using the Bioconductor package *DESeq2*. Differentially expressed gene lists were generated using the union of the genes with FDR < 0.05 and log fold change > 1 in comparison with any of the other three sample groups (i.e. *Pers1* versus *PAt, Arrested* and CD44$^{hi}$). PCA was performed using the CRAN package *Stats*, and the 3D plot was generated by the CRAN package *rgl*. Heatmap was generated by the CRAN package *gplots*, and hierarchical clustering used a complete linkage algorithm. Pathway analysis was done using the Bioconductor package *clusterProfiler* and the Reactome database through the package *ReactomePA*.

## Timer data analysis

Sample data including a negative control were batch gated for T-cell populations and exported by FlowJo (FlowJo LLC, OR) into csv files, including all compensated fluorescence data in the fcs file. The code developed in this study imports csv files into R, pre-processes and normalises data, and automatically identifies Timer$^+$ cells and performs trigonometric data transformation, producing Timer-Angle and Timer-Intensity data for individual cells in each sample, as previously described (Bending *et al*, 2018). Basic procedures for flow cytometric data analysis have been previously described elsewhere (Fujii *et al*, 2016).

## Statistical analysis and data visualisation

Statistical analysis was performed on R or Prism 6 (GraphPad) software. Percentage data for Timer$^+$ and Timer locus analysis were analysed by Mann–Whitney *U*-test or Kruskal–Wallis test with Dunn's multiple comparisons using the CRAN package *PMCMR*. Samples with fewer than 20 Timer$^+$ cells were not included in the analysis (except for Fig 7G, where threshold was set as 10). For analysis of mRNA, ear thickness data, and Timer-Angle and intensity, Student's *t*-test was used for comparison of two means. For

comparison of more than two means, a one-way ANOVA with Tukey's post hoc test was applied using the CRAN package *Stats*. For comparison of variation between two data sets across two variables, a two-way ANOVA was used with Sidak's multiple comparisons test. Scatter plots and density plots were produced by the CRAN packages *ggplot2* and *graphics*. All computations were performed on Mac (version 10.11.6). Adobe Illustrator (CS5) was used for compiling figures and designing schematic figures. Variance is reported as SD or SEM unless otherwise stated. $*P < 0.05$, $**P < 0.01$, $***P < 0.001$, $****P < 0.0001$.

**Data and code availability**

RNA-seq data can be obtained at NCBI GEO with the accession number GSE89481. All R codes are available upon request. Data will be made available upon reasonable requests to the corresponding author.

**Expanded View** for this article is available online.

## Acknowledgements

We thank for their kind support at the Flow Cytometry facility, Dr. Ayad Eddaoudi and Ms Stephanie Canning (University College London), and also Ms Jane Srivastava, Ms Catherine Simpson, and Ms Jess Rowley (Imperial College London). M.O. is a David Phillips Fellow (BB/J013951/2) from the Biotechnology and Biological Sciences Research Council (BBSRC). T.C. is supported by Great Ormond Street Hospital Children's Charity and the National Institute of Health Biomedical Research Centre at Great Ormond Street Hospital for Children NHS Foundation Trust and University College London. A.P holds a BBSRC studentship, and C.D. holds a Medical Research Council studentship. We thank Dr. Miho Ishida (University College London) for technical advice and help.

## Author contributions

MO conceived the Tocky strategies and the project. DB, TC and MO conceived and designed immunological experiments. DB, AP, CD and PPM performed animal experiments. DB and AP performed molecular experiments. DB performed RNA-seq experiment. MO wrote computational codes, and performed bioinformatics analysis and data visualisation. DB, TC and MO wrote the manuscript.

## Conflict of interest

The authors declare that they have no conflict of interest.

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
