## [Review Process File · The EMBO Journal]

A temporally dynamic Foxp3 autoregulatory transcriptional circuit controls the effector Treg programme

David Bending, Alina Paduraru, Catherine Ducker, Paz Prieto Martín, Tessa Crompton, and Masahiro Ono

Review timeline:

Submission date:	11th Jan 18
Editor Correspondence	2nd March 2018
Editorial Decision:	7th Mar 18
Revision received:	15th May 18
Accepted:	12th Jun 18

Editor: Karin Dumstrei

Transaction Report:

Editor Correspondence

2nd March 2018

Thank you for submitting your manuscript to The EMBO Journal. I am sorry for the slight delay in getting back to you with the decision but I have now received the needed input.

Your manuscript has been reviewed by two referees and their comments are provided below. As you can see from the comments, the referees express interest in the study, but also raise important concerns regarding the timer system used.

As many concerns are raised and as it is unclear if they can be resolved, I would like to ask you for a point-by-point response to see what can be done to address the concerns raised within a 3-6 months timeframe before taking a decision on the paper.

There is also the issue that the validation of the fluorescein timer protein approach is not yet published in a peer-reviewed journal, which makes it more difficult to assess how good the system is. Could you let me know what the status of that manuscript is?

You can send me the point-by-point response by email. Let me know if we need to discuss anything further

REFeree REPORTS

Referee #1

'Fluorescent timer proteins' are capable of altering their emission spectrum over time, and have been used to correlate age and cellular position of tagged proteins. The timer protein used in the present study first emits a blue fluorescence and with a half-life of ~ 4 h converts into a form that produce a mature chromophore that emits red fluorescence (Red), which has a decay rate which is longer but very unfortunately not specified (Figure 1D). In ideal pulse-like transcription, by plotting blue fluorescence against red fluorescence, it is possible to identify first Blue+Red- cells which are in the process of transcribing the loci in which the timer protein has been knock-in. Those cells later acquire a Blue+Red+ status in the case transcription persists, prior to evolving into Blue-Red+ cells if transcription diminishes or stops.

The basis and validation of this system is described in a non peer reviewed paper that can be found in bioRxiv 10.1101/217687. Using the Foxp3-Tocky mice described in bioRxiv 10.1101/217687 and in which a fluorescent protein timer ('Timer') has been introduced via BAC retrofitting and subsequent transgenesis, the authors showed that Timer transcript dynamics in Foxp3+ cells correlated to that of Foxp3 transcripts. Timer-Blue fluorescence reports real-time Foxp3 transcription, while Timer-Blue and -Red fluorescence indicates ongoing transcription of

the Foxp3 locus, a point documented in Figure 2 using thymic CD4 SP cells and splenic CD4 cells. Using a oxazolone-induced hypersensitivity (CHS) model, the authors showed that a small fraction of the CD4 T cells found in the dLNs after challenge were Timer+ cells and that among the Timer+ cells most were Red+ and that they expressed various levels of Blue fluorescence.

The few CD4 T cells present in the ear skin render the interpretation of the presented data difficult (Figure 3B). The authors also defined a Timer-Angle between Persistent - Arrested loci that is supposed to reflect the recent frequency of Foxp3 transcriptional activity (but see Specific Comments). The flux from Foxp3- to Foxp3+ T cells is increased in the dLNs draining sites of inflammation, and skin-infiltrating Foxp3+ T cells might have a more active Foxp3 transcription and be short-lived. By introducing the Foxp3-Tocky BAC transgene onto mice with a Foxp3Egfp/Scurfy genotype, due to random X inactivation, it is possible to obtain CD4 T cells that all express the Foxp3-driven Timer protein, half of which will express Foxp3/EGFP and the other half express the Scurfy Foxp3 mutant protein. Those mice were sensitised with oxazolone and Timer-Blue vs Timer-Red dot plots analysed 5 days later. Foxp3 protein was found required for controlling Foxp3 transcription in a sustained manner. Timer+ CD4 T cells from Foxp3-Tocky mice sensitised with oxazolone were sorted into Persistent 1 (but see Specific Comments), Persistent 2 and Arrested fractions and each of the fraction subjected to RNA-seq. Persistent-specific genes were enriched with cell cycle-related pathways, whereas Arrested-specific genes were enriched with cytokine-signalling pathways.

Further analysis of such subsets with a panel of antibodies confirmed that the expression levels of several markers markedly declined in Arrested Foxp3 expressors. Finally, treatment with anti-TNFR11 antibody during the challenge phase of the oxazolone-based model had a very subtle effect on the course of inflammation whereas anti-OX40 treatment had a subtle effect on the course of inflammation. In conclusion, several of the views reached with the Foxp3-Tocky mouse model are more suggestive than conclusive (see Specific comments). Owing to the fact that the 'real life' Timer-Blue vs Timer-Red dot plots are quite distinct from the theoretical ones and that the gating strategy split a continuum of cells (see Figure 5), the authors need absolutely to show that the gating strategy used to define Persistent 1, Persistent 2 and Arrested fractions in Figure 5A and 7I is done in an unsupervised manner and not in a manual manner.

Specific comments.

1/ Figure 1A. The time unit is neither specified in the figure nor in the corresponding legends. What is the meaning of raw Ct values ?

2/ Figure 1B. The x and y axes need to be labeled 'Timer-Blue' and 'Timer-Red fluorescence'.

3/ In the ideal situation shown in Figure 2D, it is easy to understand the way the Timer angle is defined. However in 'real life' the situation is far more complicated even by relying on kernel-density estimation. For instance in Figure 3B, the dots form clouds that do not intersect the y and x axes at 0 (as shown in the theoretical drawing shown in Figure 3D) in 2 of the 3 panels. Therefore I am quite confused : how an angle as defined in 3D can be calculated ? Likewise, the Timer-Blue vs Timer-Red dot plot shown in Figure 5A and the gating strategy based on it is quite distinct from the ideal view provided in Figure 3D.

4/The authors refer to their bioRxiv 10.1101/217687 to define the 'Fluorescent timer protein' they used. In this bioRxiv paper, the FT protein used is referred to as 'timer' and the corresponding reference that is given is F. V. Subach et al. Nat Chem Biol 5, 118-126 (2009). In that last paper, there is 3 FT proteins (purified fast-FT, medium-FT and slow-FT). It is thus very difficult to figure out the FT protein used in the present paper.

5/ In the case, 'arrested' T cells reinitiate transcription they will merge into the 'persistent' area. It is thus extremely unfortunate for fully interpreting the present data that they can not provide a half-life for the Red timer. Is the same issue applying to the more tractable Nr4a3 (Nor1)-model shown in bioRxiv 10.1101/217687.

6/ Is there any reason for which the data based on the oxazolone model have been obtained either after the sensitisation phase (Figure 4) or after the challenge phase (Figure 3) ?

7/ In Figure 5, why Pers1 are not called New (see Figure 3D) ?

Referee #2

Bending et al. studied the temporal translational regulation of Foxp3 in vivo, using BAC Foxp3 Tocky mice, in which the newly synthesized Foxp3 protein is marked by blue color that spontaneously changes to red with a half life of 4.1 hour. Using this system, they first confirmed thymus Foxp3+ T cells consists of blue only (30%) and blue+ red+ (70%) T cells. On the other hand, spleen T cells contained a small number of blue only T cells and the majority are Blue+ Red+ (60%) or Red only (40%). The result indicates thymus actually induces new transcription of the Foxp3 gene. Since the half life of the Blue color is very short, the large fraction of spleen Blue+ Red+ cells are most likely generated in the periphery although authors did not discuss this point explicitly.

The authors applied this system to monitor Foxp3 expression in a skin allergy model induced by oxazolone. They could clearly demonstrate oxazolone that induced local inflammation induces Foxp3 positive Treg at the skin site. Skin Foxp3+ T cells consist of a whole range of Treg cell spectrum starting from blue only population to cells containing both blue and red Foxp3 proteins. The authors divided the Foxp3 Treg cells by the ratio of blue and red, and those which express both blue and red at similar levels were classified as the persistent Treg population. This population is most abundant in skin compare to draining lymph node, again suggesting the inflammatory site generates new Foxp3 positive Treg cells.

Subsequently, the authors used this system to demonstrate Foxp3 expression itself positively regulate its own transcription using the cross with scurfy transgenic mice. In the presence of scurfy which destroys the Foxp3 function, the newly synthesized Foxp3 (blue) is severely reduced while the red only population increased. Although the total Foxp3+ (colored) fraction is unchanged, suppression of IL-2 messenger RNA expression is compromised in scurfy. The results suggest the Treg function is dependent on the persistent Treg which expresses Foxp3 protein continuously.

They have also done very extensive RNA-Seq of Treg populations at various stages which can be classified by the ratio of blue and red. They can clearly distinguish the sub population of Treg by this color ratio that clearly correlates with unique groups of gene expression. Especially, persistent Treg population expresses a clear group of genes already defined for the typical Treg cells. The population which expresses Foxp3 but stopped new production of Foxp3 is at the dormant stage and may be nonfunctional.

Finally, the authors applied this system to monitor dynamics of Foxp3 expression with the physiological response in allergic sites. To show this, they employed injection of monoclonal antibody against TNFR11 or OX40. TNFR11 antibody did not affect the profile of Foxp3 expression monitored by the Tocky system. However, OX40 agonistic antibody significantly reduced the fraction of the Foxp3 persistent population although Foxp3+ new (blue) fraction is not changed. This treatment may induce either specific depletion or some sort of cell death. Consistently the allergy phenotype is enhanced by anti-OX40 treatment.

Overall this paper contains a very novel finding using a unique monitoring system of Foxp3 expression. They provided very original analysis in such a competitive field. Although the authors did not stress it is most likely Treg induction frequently takes place in the periphery in addition to thymus. All data strong suggest it is the case. Although this question is not a major argument of this paper, the authors can discuss this possibility in discussion. I strongly recommend the acceptance of this manuscript with minor modifications.

Major comments:

1. The argument of depletion or apoptosis of Treg cells can be strengthened if they monitor apoptosis markers. This comment applies to Fig3 and Fig7.
2. Arrested and PAt populations show a similar gene expression profile but distinct from the rest. It might be helpful to discuss their physiological function. Are they related with so called ex-Treg?

Minor comments:

1. FACS profile in Figure 3B at ear skin should be replaced with other individual data which have more T cells.
2. In the reference of Figure 1B, axis indication is missing.
3. In Figure 3G, PTt should be read PAt.
4. The legend of Supplementary Fig.1 seems missing. What was used for the expression level of Foxp3 in x axis?

1st Editorial Decision

7th Mar 18

Thank you for submitting your point-by-point response. I have now had a chance to take a look at it and I appreciate your response. I would therefore like to invite you to submit a revised manuscript. Please note that the revised manuscript will be sent back to the original two referees and that their support of the manuscript is important for consideration here.

Could you also make sure that the manuscript is as readable as possible and that the description of the system and the findings is clear. I think you are doing a great job in the writing but it is a complicated system used and so clarity is important.

1st Revision - authors' response

15th May 18

Response to reviewers

Referee #1

'Fluorescent timer proteins' are capable of altering their emission spectrum over time, and have been used to correlate age and cellular position of tagged proteins. The timer protein used in the present study first emits a blue fluorescence and with a half-life of ~ 4 h converts into a form that produce a mature chromophore that emits red fluorescence (Red), which has a decay rate which is longer but very unfortunately not specified (Figure 1D).

Response: We have investigated the degradation of red proteins using the Nr4a3-Tocky mouse (Bending *et al.* (2018). 'A Timer for analyzing temporally dynamic changes in transcription during differentiation in vivo. *J Cell Biol. In press*). It is usually very difficult to perform degradation assays over a period of several days because of the toxicity of protein translation inhibitors. However, taking advantage of the Nr4a3-Tocky system, which induces Timer expression only in response to TCR signals, we sorted purified Blue-Red+ CD4+ T-cells from Nr4a3-Tocky mice, and cultured them over a week in the absence of TCR signals. The half-life of Timer-Red proteins was ~120 hours by two independent experiments (see new Fig. 1E-F).

In ideal pulse-like transcription, by plotting blue fluorescence against red fluorescence, it is possible to identify first Blue+Red- cells which are in the process of transcribing the loci in which the timer protein has been knock-in. Those cells later acquire a Blue+Red+ status in the case transcription persists, prior to evolving into Blue-Red+ cells if transcription diminishes or stops.

The basis and validation of this system is described in a non peer reviewed paper that can be found in bioRxiv 10.1101/217687.

Response: This paper is now in press at the *Journal of Cell Biology* as a Tool article ((Bending *et al.* (2018). 'A Timer for analyzing temporally dynamic changes in transcription during differentiation in vivo. *J Cell Biol. In press*). For the reviewers' benefit, we attach the accepted manuscript.

Using the Foxp3-Tocky mice described in bioRxiv 10.1101/217687 and in which a fluorescent protein timer ('Timer') has been introduced via BAC retrofitting and subsequent transgenesis, the authors showed that Timer transcript dynamics in Foxp3+ cells correlated to that of Foxp3 transcripts. Timer-Blue fluorescence reports real-time Foxp3 transcription, while Timer-Blue and -Red fluorescence indicates ongoing transcription of the Foxp3 locus, a point documented in Figure 2 using thymic CD4 SP cells and splenic CD4 cells. Using an oxazolone-induced hypersensitivity (CHS) model, the authors showed that a small fraction of the CD4 T cells found in the dLNs after challenge were Timer+ cells and that among the Timer+ cells most were Red+ and that they expressed various levels of Blue fluorescence.

The few CD4 T cells present in the ear skin render the interpretation of the presented data difficult (Figure 3B).

Response: We have replaced **Fig 3** with an entirely new figure and data set, which is drawn from four independent experiments. Here we show the T-cell response 5 days after sensitisation and 48 hours after the challenge phase, when cell numbers within the ears are higher. This figure represents the analysis of ear skin from 12 individual mice, see **Fig. 3I**, and representative plot **Fig. 3D**.

The authors also defined a Timer-Angle between Persistent - Arrested loci that is supposed to reflect the recent frequency of Foxp3 transcriptional activity (but see Specific Comments). The flux from Foxp3- to Foxp3+ T cells is increased in the dLNs draining sites of inflammation, and skin-infiltrating Foxp3+ T cells might have a more active Foxp3 transcription and be short-lived. By introducing the Foxp3-Tocky BAC transgene onto mice with a Foxp3Egfp/Scurfy genotype, due to random X inactivation, it is possible to obtain CD4 T cells that all express the Foxp3-driven Timer protein, half of which will express Foxp3/EGFP and the other half express the Scurfy Foxp3 mutant protein. Those mice were sensitised with oxazolone and Timer-Blue vs Timer-Red dot plots analysed 5 days later. Foxp3 protein was found required for controlling Foxp3 transcription in a sustained manner. Timer+ CD4 T cells from Foxp3-Tocky mice sensitised with oxazolone were sorted into Persistent 1 (but see Specific Comments), Persistent 2 and Arrested fractions and each of the fraction subjected to RNA-seq. Persistent-specific genes were enriched with cell cycle-related pathways, whereas Arrested-specific genes were enriched with cytokine-signalling pathways.

Response: We are pleased that the reviewer appreciates the key molecular mechanism proposed.

Further analysis of such subsets with a panel of antibodies confirmed that the expression levels of several markers markedly declined in Arrested Foxp3 expressors. Finally, treatment with anti-TNFR2 antibody during the challenge phase of the oxazolone-based model had a very subtle effect on the course of inflammation whereas anti-OX40 treatment had a subtle effect on the course of inflammation. In conclusion, several of the views reached with the Foxp3-Tocky mouse model are more suggestive than conclusive (see Specific comments). Owing to the fact that the 'real life' Timer-Blue vs Timer-Red dot plots are quite distinct from the theoretical ones

and that the gating strategy split a continuum of cells (see Figure 5), the authors need absolutely to show that the gating strategy used to define Persistent 1, Persistent 2 and Arrested fractions in Figure 5A and 7I is done in an unsupervised manner and not in a manual manner.

Response: We apologise if this was unclear, but the Persistent 1/ 2 cells in Figure 5A were not gated cells but flow-sorted ones (i.e. flow cytometric sorter was used to sort these cells, and a small proportion of the sorted cells were analysed by the flow cytometric sorter and the acquired data are shown in Figure 5A). To our knowledge, currently, no flow cytometer is equipped with the function to use derivative parameters, which makes it impossible to sort cells according to Timer angle. Therefore, in order to obtain RNA samples for the RNA-seq (Fig 5), we sorted cells according to their positions in the Blue-Red plane, and by analysing a small part of these sorted cells, we experimentally confirmed that distinct Timer populations were sorted according to Timer angle (Fig. 5D). We have added a section of text to make this more clear (page 13). In all the other figures, Timer loci were automatically identified in an entirely unsupervised manner, which is detailed in the accepted J Cell Biol paper. Data in Figure 7 are thus analysed and binned by the Timer algorithms. For clarity, we have also added a summary of the main principles of Timer data analysis into Fig. 2B.

Specific comments.

1/ Figure 1A. The time unit is neither specified in the figure nor in the corresponding legends. What is the meaning of raw Ct values ?

Response: We apologise for not specifying the time unit, which is hours. Ct value means "threshold cycle", which is recently more often used as a measurement of quantity in qPCR practice (e.g. single cell qPCR applications and Amaral et al, EMBO J, 2016).

2/ Figure 1B. The x and y axes need to be labeled 'Timer-Blue' and 'Timer-Red fluorescence'.

Response: We apologise, we have included the labelling. Importantly, we show Timer-Blue in y-axis so that Timer-Angle proceeds in a clock-wise manner.

3/ In the ideal situation shown in Figure 2D, it is easy to understand the way the Timer angle is defined. However in 'real life' the situation is far more complicated even by relying on kernel-density estimation. For instance in Figure 3B, the dots form clouds that do not intersect the y and x axes at 0 (as shown in the theoretical drawing shown in Figure 3D) in 2 of the 3 panels. Therefore I am quite confused : how an angle as defined in 3D can be calculated ? Likewise, the Timer-Blue vs Timer-Red dot plot shown in Figure 5A and the gating strategy based on it is quite distinct from the ideal view provided in Figure 3D.

Response: The Timer data analysis is a data pre-processing method and does not include any assumptions on data distribution. The algorithm for defining Timer angle is detailed in (Bending *et al.* (2018). 'A Timer for analyzing temporally dynamic changes in transcription during differentiation in vivo. *J Cell Biol. In press*). Briefly, the thresholds for Blue and Red autofluorescence are determined in a data-oriented manner using a negative control data for Timer expression, and then data are normalised across Blue and Red fluorescence, in order to define Timer angle values. Since the Timer data analysis is not based on any assumptions on data distribution, it will be straightforward to further analyse processed data (i.e. Timer-Angle) using common analysis methods, including manual gating and clustering methods (e.g. those based on kernel density estimation). We have added a brief summary schematic to **Fig. 2B**.

4/The authors refer to their bioRxiv 10.1101/217687 to define the 'Fluorescent timer protein' they used. In this bioRxiv paper, the FT protein used is referred to as 'timer' and the corresponding reference that is given is F. V.

Subach et al. Nat Chem Biol 5, 118-126 (2009). In that last paper, there is 3 FT proteins (purified fast-FT, medium-FT and slow-FT). It is thus very difficult to figure out the FT protein used in the present paper.

Response: The protein we use is FT-Fast, and we have referenced this in the text (page 7).

5/ In the case, 'arrested' T cells reinitiate transcription they will merge into the 'persistent' area. It is thus extremely unfortunate for fully interpreting the present data that they can not provide a half-life for the Red timer. Is the same issue applying to the more tractable Nr4a3 (Nor1)-model shown in bioRxiv 10.1101/217687.

Response: We have used Nr4a3-Tocky to provide an estimate of the red half-life, as stated above (Fig. 1E-F).

6/ Is there any reason for which the data based on the oxazolone model have been obtained either after the sensitisation phase (Figure 4) or after the challenge phase (Figure 3) ?

Response: We thank the reviewer for the thoughtful insight. In the revised manuscript, we show both sensitisation and challenge phases in normal mice (Figure 3). In contrast, the challenge phase allows us to analyse the immediate response of skin-infiltrating T cells. For clarity, the new Fig. 3 now details the T-cell response during both Sensitisation and Challenge phases.

7/ In Figure 5, why Pers1 are not called New (see Figure 3D) ?

Response: Because the calculated angle values of the sorted populations were mostly in the persistent locus.

Referee #2

Bending et al. studied the temporal translational regulation of Foxp3 in vivo, using BAC Foxp3 Tocky mice, in which the newly synthesized Foxp3 protein is marked by blue color that spontaneously changes to red with a half life of 4.1 hour. Using this system, they first confirmed thymus Foxp3+ T cells consists of blue only (30%) and blue+ red+ (70%) T cells. On the other hand, spleen T cells contained a small number of blue only T cells and the majority are Blue+ Red+ (60%) or Red only (40%). The result indicates thymus actually induces new transcription of the Foxp3 gene. Since the half life of the Blue color is very short, the large fraction of spleen Blue+ Red+ cells are most likely generated in the periphery although authors did not discuss this point explicitly.

Response: We thank the reviewer for the constructive comment. We have further analysed data in Figure 2 and showed clearly that up to 0.3% of CD4+ T cells newly express Foxp3 in the periphery per 4 hours (i.e. as "New" Blue+Red- cells; Figure 2E). Furthermore, we have added to the discussion about this important point (p. 20, line 10 -).

The authors applied this system to monitor Foxp3 expression in a skin allergy model induced by oxazolone. They could clearly demonstrate oxazolone that induced local inflammation induces Foxp3 positive Treg at the skin site. Skin Foxp3+ T cells consist of a whole range of Treg cell spectrum starting from blue only population to cells containing both blue and red Foxp3 proteins. The authors divided the Foxp3 Treg cells by the ratio of blue and red, and those which express both blue and red at similar levels were classified as the persistent Treg population. This population is most abundant in skin compare to draining lymph node, again suggesting the inflammatory site generates new Foxp3 positive Treg cells.

Subsequently, the authors used this system to demonstrate Foxp3 expression itself positively regulate its own transcription using the cross with scurfy transgenic mice. In the presence of scurfy which destroys the Foxp3 function, the newly synthesized Foxp3 (blue) is severely reduced while the red only population increased.

Although the total Foxp3+ (colored) fraction is unchanged, suppression of IL-2 messenger RNA expression is compromised in scurfy. The results suggest the Treg function is dependent on the persistent Treg which expresses Foxp3 protein continuously.

Response: We are pleased that both reviewers accepted the key molecular mechanism proposed in the paper.

They have also done very extensive RNA-Seq of Treg populations at various stages which can be classified by the ratio of blue and red. They can clearly distinguish the sub population of Treg by this color ratio that clearly correlates with unique groups of gene expression. Especially, persistent Treg population expresses a clear group of genes already defined for the typical Treg cells. The population which expresses Foxp3 but stopped new production of Foxp3 is at the dormant stage and may be nonfunctional.

Response: We are pleased that the reviewer appreciates our RNA-seq analysis. We thank the reviewer for providing us the thoughtful insights into the dormant Treg population. We have improved the discussion and discussed the relationship between the dormant Treg and memory T cells (p. 22 line 3-).

Finally, the authors applied this system to monitor dynamics of Foxp3 expression with the physiological response in allergic sites. To show this, they employed injection of monoclonal antibody against TNFR11 or OX40. TNFR11 antibody did not affect the profile of Foxp3 expression monitored by the Tocky system. However, OX40 agonistic antibody significantly reduced the fraction of the Foxp3 persistent population although Foxp3+ new (blue) fraction is not changed. This treatment may induce either specific depletion or some sort of cell death. Consistently the allergy phenotype is enhanced by anti-OX40 treatment.

Response: We are pleased that the reviewer values our immunotherapy analysis.

Overall this paper contains a very novel finding using a unique monitoring system of Foxp3 expression. They provided very original analysis in such a competitive field. Although the authors did not stress it is most likely Treg induction frequently takes place in the periphery in addition to thymus. All data strong suggest it is the case. Although this question is not a major argument of this paper, the authors can discuss this possibility in discussion. I strongly recommend the acceptance of this manuscript with minor modifications.

Response: We thank the reviewer for appreciating the novelty in our work, and for their extremely positive recommendation regarding publication. We have added a section that addresses peripheral induction of Foxp3 to the discussion (p. 20, line 11-).

Major comments:

1. The argument of depletion or apoptosis of Treg cells can be strengthened if they monitor apoptosis markers. This comment applies to Fig3 and Fig7.

Response: We thank the reviewer for this constructive comment. We have used AnnexinV as a marker of early stage apoptosis, and found that OX40+ skin infiltrating CD4+ T-cells contained more AnnexinV+ cells than OX40- counterparts, and that Foxp3+ cells had more AnnexinV+ cells. Considering that Annexin V stains preapoptotic activated T cells, the data suggest that anti-OX40 accelerated the death of those OX40+Foxp3+ cells, and that Foxp3+ effector Treg in the skin are in fact preapoptotic activated cells, which function as suppressor T cells by acting as decoy cells. Further study is required to fully reveal the mechanistic basis for the effects of anti-OX40, which is beyond the scope of this manuscript, we included the finding in Fig. EV2 and discussion (page 21, line 3 -).

2. Arrested and PAt populations show a similar gene expression profile but distinct from the rest. It might be helpful to discuss their physiological function. Are they related with so called ex-Treg?

Response: We have included a section in the discussion regarding their physiological function (**Page 22 line 6 -**).

Minor comments:

1. FACS profile in Figure 3B at ear skin should be replaced with other individual data which have more T cells.

Response: We have replaced Fig. 3 entirely, see above comment to Referee 1.

2. In the reference of Figure 1B, axis indication is missing.

Response: We apologise for this oversight, and have added the axis label (hours).

3. In Figure 3G, PTt should be read PAt.

Response: Fig. 3 has been replaced.

4. The legend of Supplementary Fig.1 seems missing. What was used for the expression level of Foxp3 in x axis?

Response: These are the log of the normalised read counts, and this has been added to the legend (Now **Fig. EV1**)

Accepted

12th Jun 18

Thank you for submitting your revised manuscript to The EMBO Journal. Your study has now been re-reviewed by the two referees and their comments are provided below.

As you can see below, both referees appreciate the introduced changes and support publication here. I am therefore very pleased to accept the manuscript for publication here.

Referee #1:

The authors have carefully addressed the major concerns I raised. In addition, the fact that the related manuscript is in press in JCB will permit to give to the readers a comprehensive picture of the model.

Referee #2:

I found the revised manuscript(EMBOJ-2018-99013R) addressed my comments in a satisfactory manner.

I have no further comment and recommend acceptance of this manuscript.

Corresponding Author Name: Masahiro Ono

Manuscript Number: EMBOJ-2018-99013